# Bacterial exonuclease III expands its enzymatic activities on single-stranded DNA

Hao Wang[1], Chen Ye[1], Qi Lu[2], Zhijie Jiang[1], Chao Jiang[3], Chun Zhou[4], Na Li[1], Caiqiao Zhang[1], Guoping Zhao[5,6,7], Min Yue[1,2,5,8]*, Yan Li[1,2]*

[1]Department of Veterinary Medicine, Zhejiang University College of Animal Sciences, Hangzhou, China; [2]Hainan Institute of Zhejiang University, Sanya, China; [3]Life Sciences Institute, Zhejiang University, Hangzhou, Zhejiang, China; [4]School of Public Health, and Department of Pathology of Sir Run Run Shaw Hospital, Zhejiang University School of Medicine, Hangzhou, China; [5]Key Laboratory of Systems Health Science of Zhejiang Province, School of Life Science, Hangzhou Institute for Advanced Study, University of Chinese Academy of Sciences, Hangzhou, China; [6]CAS Key Laboratory of Synthetic Biology, Institute of Plant Physiology and Ecology, Shanghai Institutes for Biological Sciences, Chinese Academy of Sciences, Shanghai, China; [7]Department of Microbiology and Microbial Engineering, School of Life Sciences, Fudan University, Shanghai, China; [8]State Key Laboratory for Diagnosis and Treatment of Infectious Diseases, National Clinical Research Center for Infectious Diseases, National Medical Center for Infectious Diseases, The First Affiliated Hospital, College of Medicine, Zhejiang University, Hangzhou, China

*For correspondence: myue@zju.edu.cn (MY); yanli3@zju.edu.cn (YL)

**Competing interest:** The authors declare that no competing interests exist.

**Abstract** Bacterial exonuclease III (ExoIII), widely acknowledged for specifically targeting double-stranded DNA (dsDNA), has been documented as a DNA repair-associated nuclease with apurinic/apyrimidinic (AP)-endonuclease and 3′→5′ exonuclease activities. Due to these enzymatic properties, ExoIII has been broadly applied in molecular biosensors. Here, we demonstrate that ExoIII (*Escherichia coli*) possesses highly active enzymatic activities on ssDNA. By using a range of ssDNA fluorescence-quenching reporters and fluorophore-labeled probes coupled with mass spectrometry analysis, we found ExoIII cleaved the ssDNA at 5′-bond of phosphodiester from 3′ to 5′ end by both exonuclease and endonuclease activities. Additional point mutation analysis identified the critical residues for the ssDNase action of ExoIII and suggested the activity shared the same active center with the dsDNA-targeted activities of ExoIII. Notably, ExoIII could also digest the dsDNA structures containing 3′-end ssDNA. Considering most ExoIII-assisted molecular biosensors require the involvement of single-stranded DNA (ssDNA) or nucleic acid aptamer containing ssDNA, the activity will lead to low efficiency or false positive outcome. Our study revealed the multi-enzymatic activity and the underlying molecular mechanism of ExoIII on ssDNA, illuminating novel insights for understanding its biological roles in DNA repair and the rational design of ExoIII-ssDNA involved diagnostics.

## eLife assessment

This manuscript highlights single-stranded DNA exo- and endo-nuclease activities of ExoIII as a potential caveat and an underestimated source of decreased efficiency in its use in biosensor assays. The data present **solid** evidence for the ssDNA nuclease activity of ExoIII and identifies residues that contribute to it. The findings are **useful**, but some aspects in the study remain **incomplete**.

## Introduction

Endogenous and exogenous stresses frequently result in DNA damage. Base damage is one of the most common DNA lesions caused by alkylation, oxidation, deamination, and depurination/depyrimidination (*Robertson et al., 2009*). Such structural changes in dsDNA immediately recruit a range of downstream enzymes to the lesion locus for repair, a process known as base excision repair (BER) (*van der Veen and Tang, 2015*). ExoIII has been widely recognized as one of the most critical players in BER (*Lovett, 2011*; *Lee et al., 2022*; *Mol et al., 1995*). Based on the enzymatic activities involved, BER has been conceptually divided into the following five steps (*Centore et al., 2008*): (1) glycosylase excision on the damaged base forming the apurinic/apyrimidinic (AP) site; (2) AP endonuclease (ExoIII) incision on the phosphodiester backbone to create and extend a ssDNA gap; (3) 5′-deoxyribose phosphate (dRp) removal; (4) polymerase-directed gap filling; (5) ligase-aided nick sealing. During the BER process, ExoIII performs the dual roles of AP endonuclease and exonuclease in *E. coli* (*Rogers and Weiss, 1980*). Once the damaged bases are removed by glycosylases leaving an AP site, ExoIII first acts as an endonuclease to hydrolyze the phosphodiester bond at the 5′-end of the AP site, then as an exonuclease to degrade the downstream single-stranded DNA (ssDNA) of the duplex region from 3′ to 5′ end. These two ExoIII-dominated enzymatic activities create a proper length of ssDNA gap on dsDNA, which is required for polymerase binding at the locus to re-synthesize nucleotides (*Yoo et al., 2021*). Structural analysis revealed that the $\alpha_M$ helix of ExoIII stabilized the substrate by stretching into the major groove of dsDNA in the catalytic process *Mol et al., 1995*. Interestingly, different from the typical multifunctional enzymes, i.e., DNA polymerase (*Biertümpfel et al., 2010*), with separate active sites, the multiple activities of ExoIII on dsDNA lie in the same active site (*Lee et al., 2022*; *Mol et al., 1995*).

During the 1960s-1990s, Kornberg's laboratory demonstrated that ExoIII specifically recognizes dsDNA but is inactive to ssDNA based on an ssDNA oligo of five consecutive Ts (5′ pTpTpTpTpT 3′) (*Richardson and Kornberg, 1964*; *Richardson et al., 1964*), and other researchers showed ExoIII had limited enzymatic activity on dsDNA containing 3′-protruding ssDNA (≥4 nt) (*Henikoff, 1984*; *Hoheisel, 1993*). Hence, ExoIII was previously proposed to produce ssDNA from dsDNA (*Smith, 1979*). Combined with novel materials, chemicals, or mechanical platforms, ExoIII has been widely employed in detecting nucleic acids, metals, toxins, and other types of small molecules (*Liu et al., 2021*). Based on the specific enzymatic activities of ExoIII on dsDNA in detections, the ExoIII-assisted biosensors can be divided into two categories: AP-endonuclease and exonuclease-based methods. As a promising replacement for PCR technique (*Stringer et al., 2018*), TwistAmp Liquid exo kit (https://www.twistdx.co.uk/product/twistamp-liquid-exo/) represents one of the most successful AP-endonuclease-based nucleic acid detection, which combines recombinase polymerase amplification (RPA) (*Piepenburg et al., 2006*) and ExoIII. In the application of the commercial kit, a 46~52 nt ssDNA probe is required to base-pair with the target sequence and the probe should also contain an AP site (usually simulated by a tetrahydrofuran [THF] residue) labeled with a fluorophore and a quencher, respectively, at or near the 5′- and 3′- end of AP. Once the ssDNA probe binds to the complementary sequence with the aid of recombinase and forms a dsDNA region containing an AP, ExoIII recognizes and cleaves the AP site, which separates the fluorophore and quencher, leading to a fluorescence release. In the exonuclease-based methods, ssDNA or nucleic acid aptamer contained ssDNA is often configured to form dsDNA substrate for the exonuclease activity of ExoIII (*Liu et al., 2021*). However, the digesting of ExoIII on ssDNA probe has been inadvertently observed in three ExoIII-developed detection methods (*Cai et al., 2014*; *Xu et al., 2012*; *Shen et al., 2023*) and a biochemical study (*Yang et al., 2007*). As the studies displayed limitations in proving the ssDNase activity of ExoIII: (a) no additional procedure safeguards the ssDNA substrate from forming a secondary structure containing dsDNA duplex in or between ssDNA, which can easily happen, and (b) only stays on the phenomenon without further in-depth characterization and mechanism exploring, ExoIII has, therefore, still been publicly described (https://www.sciencedirect.com/topics/biochemistry-genetics-and-molecular-biology/exonuclease-III), studied (*Lovett, 2011*; *Lee et al., 2022*; *Yoo et al., 2021*; *Shevelev and Hübscher, 2002*), and commercialized as a dsDNA-specific enzyme that is inactive or has limited activity on ssDNA (https://www.thermofisher.cn/order/catalog/product/EN0191 and https://international.neb.com/products/m0206-exonuclease-iii-e-coli#Product%20Information) for over a half-century. The definition of the enzymatic activity of ExoIII on ssDNA remains ambiguous.

To address this long-standing controversy, we used several sensitive fluorescence-quenching (FQ) ssDNA reporters to detect the enzymatic activity of ExoIII on ssDNA. FQ is distance-dependent, where increasing the distance between the fluorophore and the quencher increases the fluorescence intensity. As a sensitive and visible physical phenomenon, it has been transformed into one of the most influential techniques for monitoring dynamic alterations of macromolecule conformation in complex interactions (*Mátyus et al., 2006*; *Zhuang et al., 2000*; *Kasry et al., 2012*). The DNA or RNA-constructed FQ reporter has been successfully employed in CRISPR-based diagnostics (*Kaminski et al., 2021*; *Wang et al., 2024*) as an efficient sensor of collateral trans-cleavage activity of Cas12a on ssDNA (*Chen et al., 2018*) and Cas13a on RNA (*Gootenberg et al., 2017*). Once the nuclease cuts the ssDNA or RNA connector and disconnects the fluorophore and quencher, fluorescence will be significantly released as a positive readout for the detection. Combining with mass spectrometry and structure analysis, our finding demonstrated the ssDNase activity of ExoIII and its potential biological role in DNA repair. Furthermore, it also alerted a risk of false positive or low efficiency within the widely adopted ExoIII-ssDNA combined diagnostic approach.

## Results

### ExoIII rapidly cleaved the ssDNA FQ reporters

As an ultrasensitive biosensor of the ssDNase activity, the ssDNA FQ (*Figure 1A*) (5 nt) of CRISPR-based detection systems (*Kaminski et al., 2021*) were utilized for assaying the ssDNase activity of ExoIII and its digesting rate. To examine if the substrates form the dsDNA region internally or between ssDNA strands, the dsDNA-specific nuclease T7 exo (*Mitsunobu et al., 2014*) was used as the negative control, while LbCas12a (*Chen et al., 2018*) was designated as the positive control. Once the ssDNA FQ reporter is cleaved by nuclease, the fluorescent signal will be released. In contrast to T7 exo and LbCas12a, ExoIII (5 U/µl) triggered a substantial fluorescent signal promptly upon adding to FQ reporter (5 µM) (a total of 50 pmol), and most of the ssDNA substrate (over 90%) was cleaved in the first minute (*Figure 1B*), which suggested that the commercial ExoIII (5 U/µl) degraded over $2.7\times10^{13}$ (45 pmol) phosphodiester bonds per minute in the reaction, significantly faster than 0.7 pmol phosphodiester bonds per minute of LbCas12a and that of other nucleases (*Figure 1C*). Strikingly, we found even at room temperature (22 °C), the fluorescence release initiated immediately when 0.5 µl ExoIII was mixed with FQ reporter (5 µM) under LED blue light (*Figure 1—video 1*). In summary, T7 exo showed no significance with D-T7 exo; APE1 exhibited a significantly detectable fluorescence signal compared to its deactivated counterpart (D-APE1), albeit much weaker than that of ExoIII and LbCas12a (*Figure 1C*).

Given that ExoIII has long been recognized as a dsDNA-specific nuclease, we initially speculated that the ssDNA activity on FQ reporters might be ascribed to its cleavage at the 5′ or 3′ end phosphodiester bonds of ssDNA reporter rather than the typical phosphodiester bonds within ssDNA. To narrow down the possibility, we first removed the 5′ end phosphodiester by designing the FQ reporter with FAM labeled at its 5′ first thymine ($T_1$) base instead of the 5′ end phosphate (*Figure 1D*). Logically, the 5′ FAM $T_1$-labeled reporter should not produce fluorescence if ExoIII cuts at the 5′ end phosphodiester bond. But fluorescence monitoring assays showed that ExoIII (5 U/µl) digested over 90% of 5′ FAM $T_1$-labeled FQ reporter (5 µM) in the first minute (*Figure 1E*), indicating a speed of $2.7\times10^{13}$ (45 pmol) phosphodiester bonds per minute (*Figure 1F*); APE1 and LbCas12a also exhibited a stronger fluorescence intensity compared to their deactivated counterparts (D-APE1 and D-LbCas12a) (*Figure 1F*). Treated with ExoIII, the fluorescence of the FQ reporter was generated rapidly even before incubating at 37 °C (*Figure 1—video 2*). The evidence implied that ExoIII cleaved the FQ reporter not at 5′ end phosphodiesters.

Subsequently, we removed the 3′ end phosphodiesters by labeling the BHQ1 at the reporter's 3′ end thymine (*Figure 1G*). The fluorescence monitoring assay showed that the ExoIII-treated FQ reporter (5 µM) released a significant fluorescence signal compared to other nucleases. Slower than the previous two FQ reporters, it cleaved over 90% of the base-labeled FQ reporter in 10 min (*Figure 1H*), demonstrating that the commercial ExoIII (5 U/µl) digested about $2.7\times10^{12}$ (4.5 pmol) phosphodiester bonds or nucleotides of ssDNA in the FQ reporter (*Figure 1I*). Compared with the massive fluorescence induced by ExoIII, the modifications on thymine may impede the binding and cleaving of the LbCas12a-crRNA complex on the ssDNA substrate, leading to less fluorescence generated

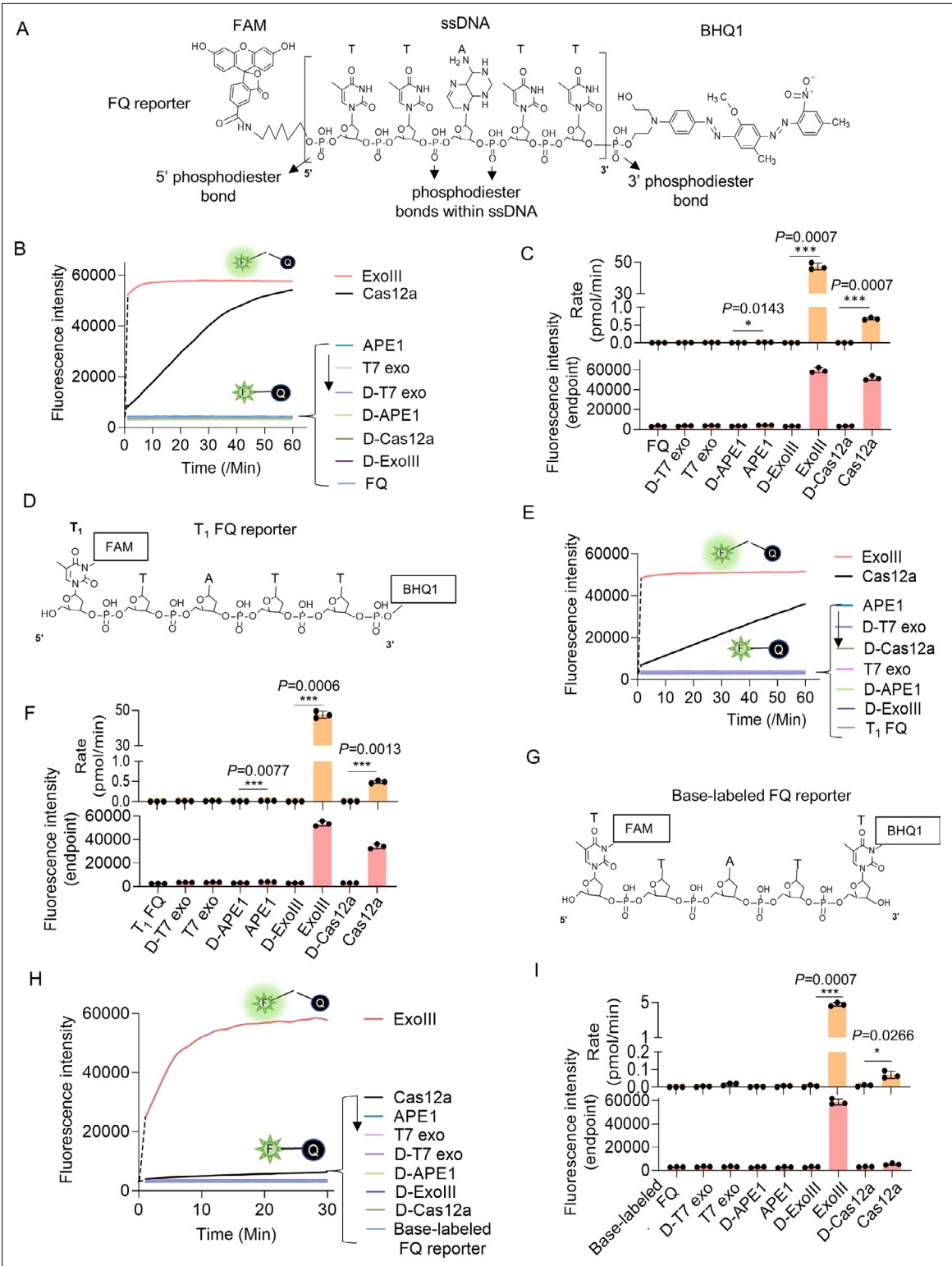

**Figure 1.** Exexonuclease Ⅲ (ExoⅢ) (5 U/µl) rapidly cleaved the ssDNA fluorescence-quenching (FQ) reporter (5 µM) of the CRISPR/Cas12a system with an average rate of 4.5~45 pmol (2.7~27 × 10¹²) phosphodiester bonds per minute. (**A**) The typical structure of the ssDNA FQ is presented, in which three types of putative susceptible sites for ExoⅢ digestion are outlined, including 5' phosphodiester bond, phosphodiester bonds within ssDNA, and 3' phosphodiester bond. (**B**) The fluorescence intensity of FQ reporter (5 µM) treated with four active nucleases (T7 exonuclease (T7 exo) (5 U/µl), APE1 (5 U/µl), ExoⅢ (5 U/µl), and Cas12a/crRNA (0.1 µM)) or their heat-deactivated counterparts (D-T7 exo, D-APE1, D-ExoⅢ, D-Cas12a) was monitored every minute for 60 min. The dsDNA nuclease T7 exo was a negative control, also used to examine if the ssDNA substrates in the reaction formed dsDNA

*Figure 1 continued on next page*

*Figure 1 continued*

regions internally and between ssDNA strands, while LbCas12a with a trans-cleavage activity on ssDNA was a positive control. The D-nuclease served as the internal control of each active nuclease. The curve was plotted with an average value of three repeats. (**C**) The average rate of ExoIII digestion on FQ reporter was calculated by the formula: (the fluorescence produced/total fluorescence × 50 pmol)/reaction time. The total fluorescence means the fluorescence intensity generated when the input FQ reporter (50 pmol) was all cleaved. p (APE1 vs. D-APE1)=0.0143; p (ExoIII vs. D-ExoIII)=0.0007; p (Cas12a vs. D-Cas12a)=0.0007. *p<0.05, and ***p<0.001. The fluorescence intensity at the endpoint of the reaction was plotted based on three repeats, after the FQ reporter respectively treated with four active nucleases and their heat-deactivated counterpart (D-nuclease). (**D**) The 5' $T_1$ FAM-labeled FQ reporter structure is designed and presented, in which the 5' phosphodiester bond is removed and 3' phosphodiester bond retained. (**E**) Real-time monitoring on the fluorescence generated upon 5' $T_1$ FAM-labeled FQ reporter (5 µM) treated with four nucleases (T7 exonuclease (T7 exo) (5 U/µl), APE1 (5 U/µl), ExoIII (5 U/µl), and Cas12a-crRNA (0.1 µM)) or deactivated nucleases (D-T7 exo, D-APE1, D-ExoIII, D-Cas12a) was performed for 60 min. The average value of four repeats was calculated and curved. (**F**) The average rate of ExoIII digestion on $T_1$-labeled FQ reporter was calculated by the formula: (the fluorescence produced/total fluorescence × 50 pmol)/reaction time. After the FQ reporter was treated with the four nucleases or their deactivated ones, the fluorescence intensities at the reaction endpoint were measured and plotted. p (APE1 vs. D-APE1)=0.0077; p (ExoIII vs. D-ExoIII)=0.0007; p (Cas12a vs. D-Cas12a)=0.0013. (**G**) The structure of the base-labeled FQ reporter for ExoIII is diagrammed, in which both 5' and 3' phosphodiester bonds are removed. (**H**) The digestion of the four nucleases and their deactivated counterparts on the base-labeled FQ reporter was monitored for 30 min. The average value of three repeats was calculated and curved. (**I**) The average rate of ExoIII digestion on the base-labeled FQ reporter was calculated by the formula: (the fluorescence produced/total fluorescence × 50 pmol)/reaction time. After the cleavage reactions on the FQ reporter, the fluorescence intensities at the reaction endpoint were measured and plotted. p (ExoIII vs. D-ExoIII)=0.000006; p (Cas12a vs. D-Cas12a)=0.0185. The statistical analysis was performed using a two-tailed *t*-test. *p<0.05, ***p<0.001. The dashed line in the figure does not indicate the real-time fluorescence generated in the reaction but only represents a trend in the period for monitor machine to initiate ~2 min.

The online version of this article includes the following video(s) for figure 1:

**Figure 1—video 1.** Exonuclease Ⅲ (ExoIII) immediately triggers significant fluorescence at room temperature when mix with the fluorescence-quenching (FQ) reporter under LED blue light.

https://elifesciences.org/articles/95648/figures#fig1video1

**Figure 1—video 2.** Exonuclease Ⅲ (ExoIII) rapidly causes significant fluorescence at room temperature when mixed with the $T_1$ labeled fluorescence-quenching (FQ) reporter under LED blue light.

https://elifesciences.org/articles/95648/figures#fig1video2

(*Figure 1I*). Combining all the results of three types of FQ reporters, we demonstrated ExoIII might harbor a powerful ssDNase activity.

## ExoIII is an efficient 3'→5' ssDNase

To exclude the possibility of sequence-specific driven catalysis of ExoIII, we tested three ssDNA oligos with fluorophore labeled at the 5' end (5 µM) (*Figure 2A*). After 40 min of incubation with ExoIII (5 U/µl) at 37 °C, the reaction mixture of three probes was separated by gel electrophoresis. The result indicated that the ssDNA was degraded by ExoIII and LbCas12a compared with the undigested product of APE1 and T7 exo (*Figure 2B and C*). The reaction products gradually decreased in length as time prolonged (*Figure 2D*), suggesting that ExoIII shortened Probe 1 into smaller fragments gradually and rapidly from 3' to 5' end. The time course analysis on the digestion demonstrated the digesting rate of ExoIII on the ssDNA stood at 20~400 pmol nucleotides per minute in the 15 min (*Figure 2E*). Similarly, Probe 2 was processed into shorter fragments by ExoIII and LbCas12a compared to other nucleases (*Figure 2F and G*) and were digested gradually and promptly over time (*Figure 2H and I*). Probe 3 also resembled the digestion over time (*Figure 2J, K, L and M*). The gradual and successive degradation on the three ssDNA substrates in the 40 min indicated that ExoIII (5 U/µl) digested the ssDNA (5 µM) in the 3'→5' direction with an estimated average speed of approximately $1.5 \times 10^{13}$ (25 pmol) phosphodiester bonds or nucleotides per minute, and the digestion rate varied with the encountered nucleotides of ssDNA substrate. We also compared the digestion rates of ExoIII on ssDNA and dsDNA. The result showed that ExoIII required 10 min to digest the 30-nt single-stranded DNA (ssDNA) (*Figure 2—figure supplement 1A*), whereas it could digest the same sequence on double-stranded DNA (dsDNA) within 1 min (*Figure 2—figure supplement 1B*). This indicated that ExoIII digested the dsDNA at a rate at least ten times faster than ssDNA. Even not as rapid as the dsDNA, the ssDNase activity of ExoIII surpasses that of the conventional ssDNA-specific nuclease ExoI (*Shen et al., 2023*), suggesting a potential biological significance of ExoIII in bacteria related to ssDNA metabolism.

To investigate the massive disparity of ssDNase activity between the two highly similar homologs, ExoIII and APE1, we performed structure alignment and found the structure of the $\alpha_M$ helix of ExoIII

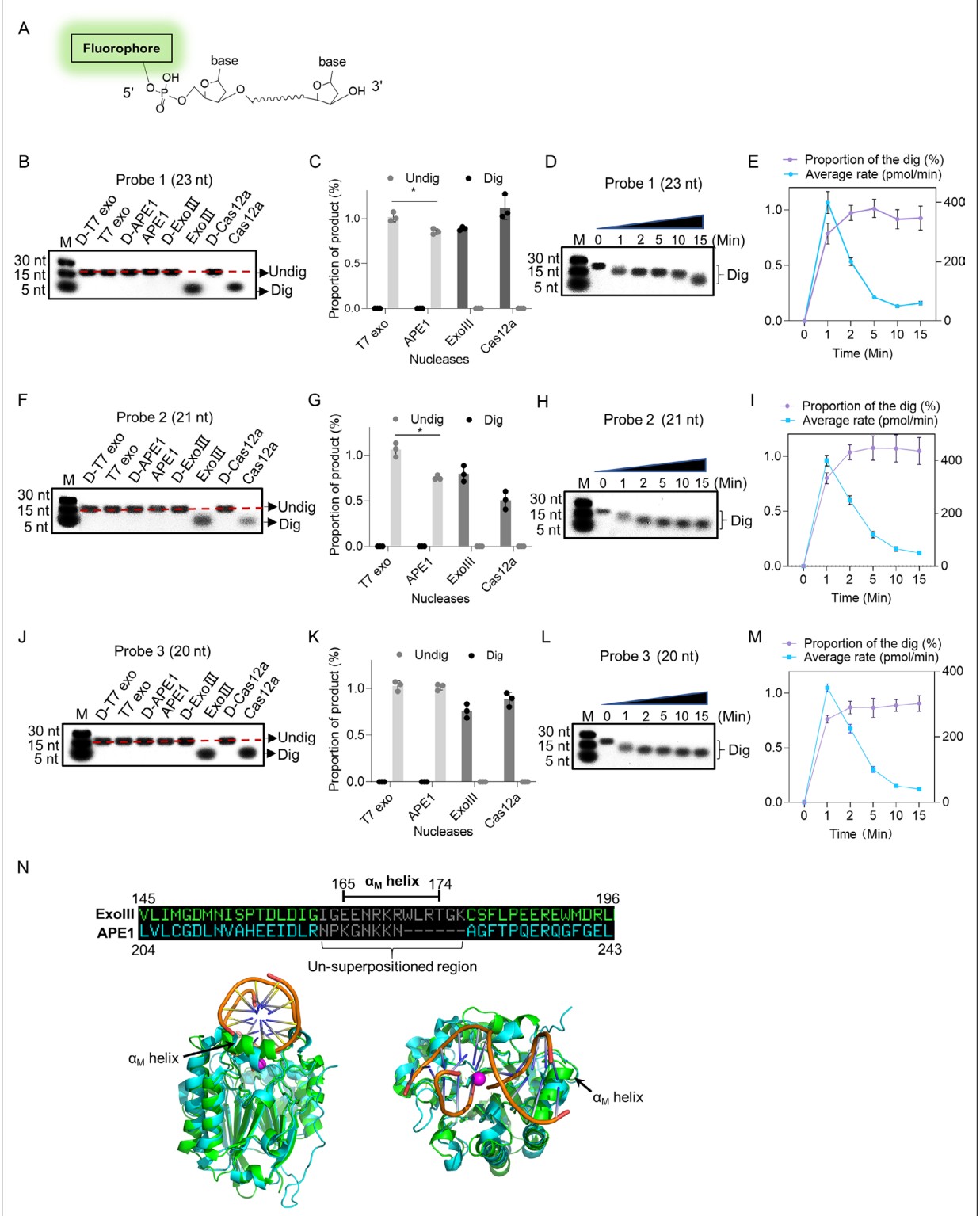

**Figure 2.** Exonuclease III (ExoIII) (5 U/μl) digested the 5′ end fluorophore-labeled ssDNA substrates (5 μM) in a 3′ to 5′-end direction with an estimated rate of ~$10^{13}$ nucleotides per minute. (**A**) The 5′ fluorophore-labeled ssDNA structure for the following ExoIII digestion is shown. The squiggle line represents the ~20 nucleotides of the ssDNA oligo. (**B, F,** and **J**) The ssDNA oligo of Probe 1, Probe 2, or Probe 3 (5 μM) was treated with the four nucleases (T7 exo (5 U/μl), APE1 (5 U/μl), ExoIII (5 U/μl), and Cas12a-crRNA (0.1 μM)) and their deactivate counterpart (D-T7 exo, D-APE1, D-ExoIII, D-Cas12a) for 30 min. The reaction products were analyzed by gel electrophoresis. Undig, undigested; dig, digested; M, maker. (**C, G,** and **K**) The gray intensity of the digested or undigested bands produced by the three probes was measured by ImageJ. The undigested or digested proportions were

*Figure 2 continued on next page*

*Figure 2 continued*

calculated with the formula: Proportion = (Intensity of the undigested or digested band produced in the active nuclease treatment)/(Intensity of the undigested band produced in the corresponding deactivated nuclease treatment). p (APE1 vs. T7 exo on Probe 1)=0.04; p (APE1 vs. T7 exo on Probe 2)=0.02; *p<0.05. The statistical analysis was performed using a two-tailed *t*-test. (**D, H**, and **L**) The time course of ExoIII digestion on the three probes was analyzed by gel electrophoresis, and the gray intensity of the generated bands was measured by ImageJ. (**E, I**, and **M**) The digesting rate of ExoIII on the ssDNA was calculated by: the digested proportion ×50 pmol × digested nucleotides/reaction time. The digested proportion was calculated by the formula: (gray intensity of the digested band produced)/(gray intensity of the band at 0 s). Detailed information on these oligos is described in *Supplementary file 2*. All experiments were repeated three times, and the representative ones were shown. (**N**) The structure alignment between ExoIII (PDB ID:1AKO) and APE1-dsDNA (PDB ID: 1DE8 and NDB ID: 5WN5) was performed by Pymol, and the un-superpositioned region (Residues 165–174, $\alpha_M$ helix) in the enzyme-substrate binding surface between ExoIII and APE1 was displayed. The aligned structures were shown from different angles by Pymol.

The online version of this article includes the following source data and figure supplement(s) for figure 2:

**Source data 1.** Exonuclease Ⅲ (ExoⅢ) rapidly digested the 5′ end fluorophore-labeled ssDNA substrates.

**Figure supplement 1.** The digestion process of exonuclease Ⅲ (ExoIII) on ssDNA and double-stranded DNA (dsDNA) substrates was compared.

**Figure supplement 1—source data 1.** The digestion process of exonuclease Ⅲ (ExoⅢ) on ssDNA and double-stranded DNA (dsDNA) substrates.

deteriorated in APE1 (*Figure 2N*). As $\alpha_M$ lies in the binding interface of ExoIII and substrate (*Mol et al., 1995*), it might play a critical role in the digestion of ssDNA. The degenerated ssDNA-cleavage activity of APE1 may evolve to minimize the risk of genomic instability in human. Whether it is associated with related human cancers remains to be further studied.

## Mass spectrometry analysis revealed the endonuclease activity of ExoIII on ssDNA

To provide high-resolution information regarding the ExoIII actions on the ssDNA substrates (used in *Figure 2*), we parsed the products of ExoIII by ESI-mass spectrometry. The major products of the three ssDNA probes are identified and listed (*Table 1*) and all the reaction products are included in *Supplementary file 1*. After being treated with ExoIII, Probe 1 was digested to the three major products (Product 1 of Probe 1, termed as P1P1, Mass peak = 2914.9 Da; P1P2, Mass peak = 2601.5 Da; P1P3, Mass peak = 5395.5 Da) (*Figure 3A*), which respectively matched to an 8-nt (Theoretical mass = 2915.6 Da), 7-nt (Theoretical mass = 2602.4 Da) and 16-nt (Theoretical mass = 5397.2 Da) ssDNA fragments of Probe 1, and these undigested nucleatides accounted for ~15% of the input (*Figure 3B*). Probe 2 was digested to two major products (P2P1, Mass peak = 4639.6 Da; P2P2, Mass peak = 957.7 Da) (*Figure 3C*), which respectively matched to a 13-nt (Theoretical mass = 4641.6 Da) and 3-nt (Theoretical mass = 957.6 Da) ssDNA fragments of Probe 2, and these remaining oligos accounted for ~0.6% of the input (*Figure 3D*). Probe 3 was digested into four products (P3P1, Mass peak = 2936.9; P3P2, Mass peak = 3240.9; P3P3, Mass peak = 2632.7; P3P4, Mass peak = 2328.7) (*Figure 3E*) respectively matched to an 8-nt (Theoretical mass = 2937.6 Da), 9-nt (Theoretical mass = 3241.8 Da), 7-nt (Theoretical mass = 2633.4 Da) and 6-nt (Theoretical mass = 2329.2 Da) fragments of Probe 3, and these leftover accounted for ~28% of the input (*Figure 3F*). According to the mass peak intensities of reaction products (*Table 2*), the majority of ssDNA substrates (5 µM) were degraded into mononucleotides (~85% for Probe 1,~99% for Probe 2 and ~72% for Probe 3) by ExoIII (5 U/µl). As the control, the mass spectrometry analysis of the three probes without ExoIII digestion is provided (*Figure 3—figure supplement 1*).

Furthermore, we found these undigested fragments above (P1P1, P1P2, P2P1, P3P1, P3P2, P3P3, and P3P4) retained an intact 5′ end fluorophore and a free 3′-OH, which indicated the ExoIII catalyzed the cleavage of ssDNA at the 5′-side bond of phosphodiester. And 17 ssDNA fragments (four fragments from Probe 1 and 13 fragments from Probe 2 included in *Supplementary file 1*) were identified from the middle part of the ssDNA, implying ExoIII possessed an endonuclease activity on ssDNA, which might be activated by specific sequences. Combined with the results of *Figure 2*, we confirmed that ExoIII is an efficient ssDNase, and its digestion rate varied based on the sequences of ssDNA.

## Reaction conditions for the ssDNase activity of ExoIII

To explore if this enzymatic action of ExoIII requires $Mg^{2+}$, we added EDTA to the enzymatic reaction. The endpoint fluorescence intensity of the FQ reporter showed that the addition of EDTA completely inhibited the fluorescence generation compared with the reaction without EDTA (*Figure 4A*). The gel

**Table 1.** The mass spectrometry analysis on the exonuclease III (ExoIII)-treated ssDNA oligos.

| Sample (Theoretical mass) | ssDNA without ExoIII | | | | ssDNA treated with ExoIII | | | | Structure of the top intensity |
|---|---|---|---|---|---|---|---|---|---|
| | Mass peak (Da) | Intensity | Relative (%) | Presumed products (Theoretical mass) | Mass peak (Da) | Intensity | Relative (%) | Presumed products (Theoretical mass) | |
| Probe 1 (7504) | 7501.1 | 4.79E+07 | 100 | FITC-CAAACCCAGAGCCAATCTTATCT (7504) | 2914.9 | 6.31E+06 | 100 | FITC-CAAACCCA (2915.6) |  |
| | 7523.4 | 1.69E+07 | 35.28 | FITC-CAAACCCAGAGCCAATCTTATCT (7504) + Na$^+$ | 2601.5 | 4.76E+06 | 75.37 | FITC-CAAACCC (2602.4) | |
| | 7546.1 | 1.02E+07 | 21.4 | FITC-CAAACCCAGAGCCAATCTTATCT (7504)+2Na$^+$ | 4047.7 | 1.10E+06 | 17.43 | Undefined | |
| | 7567.9 | 3.45E+06 | 7.2 | FITC-CAAACCCAGAGCCAATCTTATCT (7504)+3Na$^+$ | 2312.2 | 1.07E+06 | 17.01 | Undefined | |
| | 10106.2 | 3.10E+06 | 6.47 | Undefined | 5395.5 | 6.18E+05 | 9.79 | FITC-CAAACCCAGAGCCAAT (5397.2) | |
| Probe 2 (7063) | 7059.7 | 3.34E+07 | 100 | FAM-GGGTGGGCGGAAAACTATTTC (7063) | 1287.5 | 5.53E+05 | 100 | Mass peak of ExoIII |  |
| | 7082.3 | 7.84E+06 | 23.46 | FAM-GGGTGGGCGGAAAACTATTTC (7063)+Na + | 627.8 | 3.35E+05 | 60.59 | Mass peak of ExoIII or TT (P) (626.4) | |
| | 7104.6 | 2.95E+06 | 8.84 | FAM-GGGTGGGCGGAAAACTATTTC (7063)+2Na$^+$ | 13646.1 | 2.01E+05 | 36.26 | Undefined | |
| | 10106.5 | 2.27E+06 | 6.8 | Undefined | 10107.2 | 1.97E+05 | 35.57 | Undefined | |
| | 7341.3 | 1.20E+06 | 3.58 | Undefined | 1979.4 | 1.92E+05 | 34.81 | Undefined | |
| | 13461.6 | 1.07E+006 | 7.94 | Undefined | 4639.6 | 1.59E+05 | 28.74 | FAM-GGGTGGGCGGAAA (4641.6) | |
| | 11549.0 | 6.24E+005 | 6.55 | Undefined | 6655.3 | 1.55E+05 | 28.11 | Undefined | |
| | 3376.6 | 4.57E+005 | 2.88 | Undefined | 1291.6 | 1.45E+05 | 26.29 | Mass peak of ExoIII or GCGG/GGCG/GGGC (P) (1293.8) | |
| | 7481.5 | 3.82E+005 | 5.67 | Undefined | 9126.5 | 1.02E+05 | 18.42 | Undefined | |
| | | | | | 957.7 | 1.02E+05 | 18.39 | AAA (P) (957.6) | |

*Table 1 continued on next page*

Table 1 continued

| Sample (Theoretical mass) | ssDNA without ExoIII | | | | ssDNA treated with ExoIII | | | | Structure of the top intensity |
|---|---|---|---|---|---|---|---|---|---|
| | Mass peak (Da) | Intensity | Relative (%) | Presumed products (Theoretical mass) | Mass peak (Da) | Intensity | Relative (%) | Presumed products (Theoretical mass) | |
| Probe 3 (6658) | 6655.2 | 3.30E+07 | 100 | FAM-AGTCCGT TTGTTCTTGTGGC (6658) | 2936.9 | 8.36E+06 | 100 | FAM-AGTCCGTT (2937.6) |  |
| | 6677.5 | 1.06E+07 | 32.18 | FAM-AGTCCGTTTG TTCTTGTGGC (6658) + Na$^+$ | 3240.9 | 2.77E+06 | 33.1 | FAM-AGTCCGTTT (3241.8) | |
| | 6699.9 | 6.00E+06 | 18.15 | FAM-AGTCCGTTTT GTTCTTGTGGC (6658)+2Na$^+$ | 2632.7 | 2.34E+06 | 27.99 | FAM-AGTCCGT (2633.4) | |
| | 6721.6 | 1.85E+06 | 5.59 | Undefined | 2328.7 | 8.11E+05 | 9.69 | FAM-AGTCCG (2329.2) | |
| | 6936.6 | 1.29E+06 | 3.91 | Undefined | 643.8 | 4.59E+05 | 5.48 | Undefined | |
| ExoIII | | | | | 1287.6 | 5.44E+05 | 100 | Undefined | |
| | | | | | 627.8 | 3.45E+05 | 63.38 | Undefined | |
| | | | | | 2639.4 | 1.91E+05 | 35.09 | Undefined | |
| | | | | | 1291.6 | 1.47E+05 | 27.04 | Undefined | |
| | | | | | 19174.8 | 1.36E+05 | 25.08 | Undefined | |

Note that P indicates the oligo with a phosphate group. A (dAMP)=331.2 g/mol, T (dTMP)=322.2 g/mol, G (dGMP)=347.2 g/mol, and C (dCMP)=307.2 g/mol. Molecular weight of the 5′ end FITC part (C27H25N2O5S) of the reaction product is 489 g/mol, and the 5′ end FAM part (C$_{27}$H$_{24}$NO$_6$) is 458 g/mol. The bold letter indicates the zone of three identical bases where the digestion of ExoIII stalled.

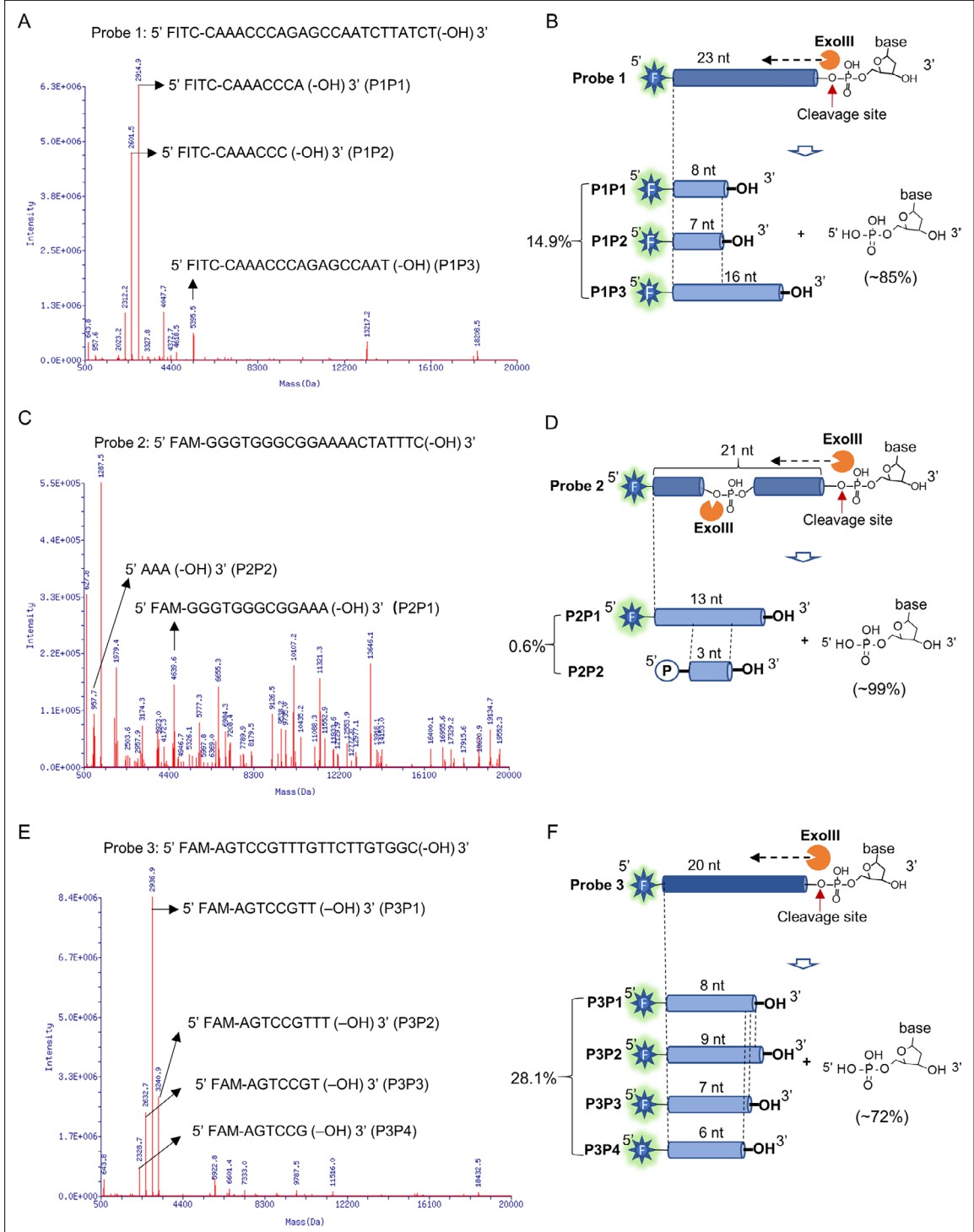

**Figure 3.** Mass spectrometry analysis revealed the exonuclease and endonuclease activities of exonuclease III (ExoIII) on ssDNA. After the ssDNA probe (5 µM) was incubated with ExoIII (5 U/µl) at 37 °C for 30 min, the reaction product was analyzed by mass spectrometry. The detection range of mass spectrometry is 500~20,000 Da. (**A**) The mass peaks of ssDNA Probe 1 digested by ExoIII are present. The three major peaks of reaction products of Probe 1 (P1P1, P1P2, and P1P3) are indicated by arrows. The information related to major peaks is provided in *Table 1*. (**B**) The digestion process, including the cleavage site, product, and relevant proportion of Probe 1 is illuminated based on the mass spectrometry analysis. The three major mass peaks match three fragments of Probe 1 (8 nt, 7 nt, and 16 nt), respectively. (**C**) The mass peaks of ssDNA Probe 2 digested by ExoIII are displayed. The two major peaks (P2P1 and P2P2) are labeled by arrows. (**D**) The digestion process of Probe 2 is illuminated based on the mass spectra analysis. The two major mass peaks of reaction products match two fragments of Probe 2 (13 nt and 3 nt). (**E**) The mass peaks of Probe 3 digested by ExoIII are displayed. The major peaks (P3P1, P3P2, P3P3, and P3P4) are labeled by arrows. (**F**) The reaction process of Probe 3 is illuminated based on the mass spectra

*Figure 3 continued on next page*

*Figure 3 continued*

analysis and gel result (**L**). The four major mass peaks of reaction products match the fragments of Probe 3 (8 nt, 9 nt, 7 nt, and 6 nt). As the control, the mass peaks of undigested ssDNA probes are displayed in *Figure 3—figure supplement 1*.

The online version of this article includes the following figure supplement(s) for figure 3:

**Figure supplement 1.** As controls of the exonuclease Ⅲ (ExoIII)-treated ssDNAs, the mass peaks of ssDNA probes undigested by ExoIII were present at (**A**) for Probe 1, (**B**) for Probe 2, and (**C**) for Probe 3.

analysis on the cleavage product of three ssDNA probes showed that adding EDTA stopped the catalytic digestion of ExoIII on ssDNA (*Figure 4B*). These EDTA-added experiments demonstrated that $Mg^{2+}$ was required for the ssDNase activity of ExoIII. Subsequently, we added different concentrations of ExoIII (0, 0.025, 0.125, 0.250, 0.500, and 1.250 µM) to the ssDNA probe (0.5 µM). The result showed the amount of ExoIII (0.025 µM) degraded ~10 nucleotides of ssDNA substrate (0.5 µM) in 10 min (*Figure 4C and D*), suggesting each molecular of ExoIII was capable of cleaving off one nucleotide by 3 s. To explore the suitable temperature for the ssDNase activity, ExoIII was incubated with the ssDNA probe at 0, 16, 25, 37, 42, and 50 °C, respectively. The gel results indicated the ssDNase activity of ExoIII was active at 16, 25, and 37 °C but receded when the temperature was raised to 42°C and 50°C (*Figure 4E and F*), suggesting the suitable temperature for the ssDNase activity of ExoIII was probably at a range of 16–42 °C.

Intriguingly, focusing on the major products of the three ssDNA probes in *Figure 3*, we found digestion of ExoIII seemingly always stalled in or ahead of a zone of serial identical bases (*Table 1*), suggesting this type of sequence feature might affect the digestion of ExoIII. To explore if this phenomenon always occurs upon serial identical bases, we performed enzymatic reactions on three 5' FAM-labeled ssDNA probes consisting of 20 nt identical nucleotides. The reaction product was analyzed by gel after these ssDNA substrates (5 µM) were incubated with purified ExoIII (2.5 µM) for 15 min. The result of the time course analysis indicated that $A_{20}$, $C_{20}$, and $T_{20}$ displayed an obviously slower degradation than the ssDNA substrates in *Figure 2* ($A_{20} \approx C_{20} > T_{20}$) (*Figure 4G and H*), suggesting the consecutive identical nucleotides in ssDNA might slow down the digestion of ExoIII.

**Table 2.** The embedded use of ssDNA in various exonuclease Ⅲ (ExoIII)-associated diagnostic platforms.

| Classification by the ExoIII activity used in the methods | Diagnostic platforms | Targets | Functions of ExoIII | Involved nucleic acid aptamer | Published |
|---|---|---|---|---|---|
| dsDNA exonuclease-based detection | ExoIII-aided electrochemiluminescence techniques | miRNA-21 (a cancer biomarker) | Recognize and degrade dsDNA to ssDNA for triggering subsequent reaction | The ssDNA oligos and hairpin dsDNA with 3' end protruding ssDNA or dsDNA with the blunt end | *Zhang et al., 2023* |
| | ExoIII combined with quantum dots. | HIV and HBV virus | | | *Wang et al., 2021* |
| | ExoIII combined with nanoparticles | p53 gene | | | *Wu et al., 2021* |
| | ExoIII integrated with the microfluidic platform. | Manganese superoxide dismutase gene | | | *Zheng et al., 2020* |
| | ExoIII facilitated chemiluminescence techniques. | Synthetic DNA target | | | *Gao and Li, 2014* |
| AP-endonuclease based detections | ExoIII integrated with Recombinase Polymerase Amplification (RPA) | Aeromonas salmonicida | Recognize and cleave the AP site on dsDNA for releasing fluorescence signal. | Primers and a ssDNA oligo containing AP site labeled with fluorescence and quencher groups | *Zhou et al., 2022* |
| | | Burkholderia cepacia | | | *Daddy Gaoh et al., 2023* |
| | | Elizabethkingia miricola | | | *Qiao et al., 2022* |
| | | Porcine parvovirus | | | *Wang et al., 2017* |
| | | Dengue virus | | | *Abd El Wahed et al., 2015* |
| | | SARS-CoV-2 | | | *Behrmann et al., 2020* |

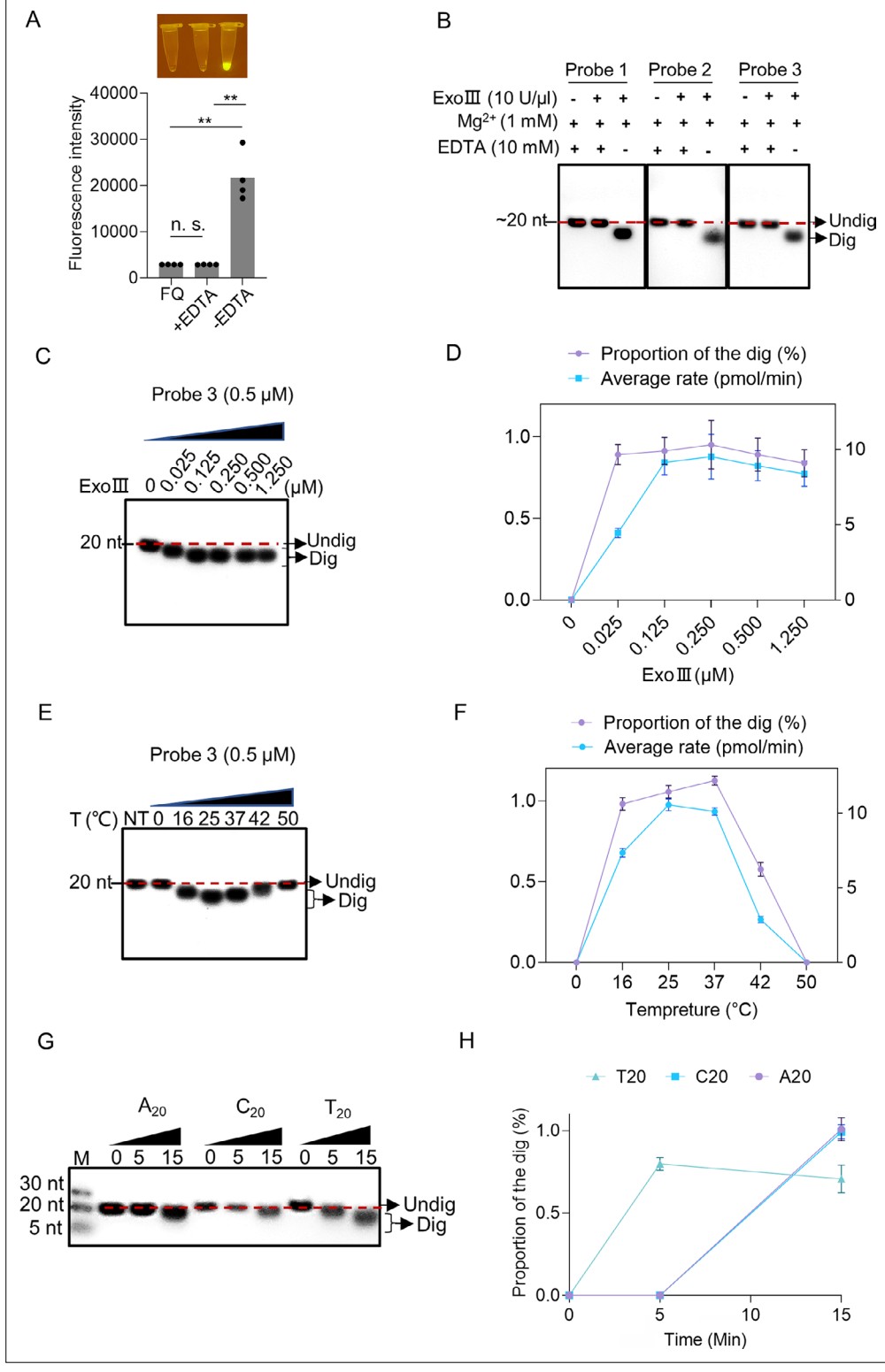

**Figure 4.** Enzymatic reaction conditions for the ssDNase activity of exonuclease III (ExoIII) were investigated.
(**A**) EDTA (10 mM) was added with the mixture of fluorescence-quenching (FQ) reporter (2.5 μM) and MgCl$_2$
(1 mM) before incubating with commercial ExoIII (5 U/μl) for 10 min. The fluorescence intensity generated in
10 min was measured and plotted on four repeats. The corresponding fluorescent tubes at the reaction endpoint
were captured under LED UV light. The reaction without adding EDTA was regarded as a control. **p<0.01. The
n.s. indicates no significance. Statistical significance was determined by a two-tailed *t*-test. (**B**) Three ssDNA

*Figure 4 continued on next page*

*Figure 4 continued*

probes (5 µM) with or without adding EDTA were digested by commercial ExoIII (5 U/µl) for 10 min, and the reaction products were analyzed by gel. (**C**) Different amounts of purified ExoIII (0, 0.025, 0.125, 0.250, 0.500, and 1.250 µM) were incubated with ssDNA Probe 1 (0.5 µM) for 10 min, and the reaction products were analyzed by gel electrophoresis. (**D**) The digesting rate of ExoIII on the ssDNA was calculated by: the digested proportion × 50 pmol × digested nucleotides/reaction time. The digested proportion of Probe 3 was calculated by: (Gray intensity of the digested band)/(Gray intensity of the undigested band with 0 µM ExoIII). (**E**) The ssDNA substrate (0.5 µM) was incubated with ExoIII (0.5 µM) at different temperatures (0, 16, 25, 37, 42, and 50 °C) for 10 min, and the products were analyzed by gel electrophoresis. NT, no treatment. (**F**) The digested proportion of Probe 3 was calculated by: (Gray intensity of the digested band)/Gray intensity of the undigested band (NT). (**G**) The 5′ FAM-labelled ssDNA probes constructed by 20 consecutive identical bases ($A_{20}$, $C_{20}$, $T_{20}$) were digested by ExoIII over 15 min, and the reaction products were separated by gel electrophoresis. The fluorescence intensity difference between the three oligos is caused by the bases covalently linked with a fluorophore. $G_{20}$ was not tested here as it easily forms a G-quadruplex structure. (**H**) The gray intensity was determined by ImageJ. A digested proportion curve over 15 min was plotted from three repeats. The digested proportion was calculated by the formula: (Intensity of the digested band produced)/(Intensity of the band at 0 s). Data represents the average value of three repeats and is expressed as mean ± SD.

The online version of this article includes the following source data for figure 4:

**Source data 1.** Enzymatic reaction conditions for the ssDNase activity of exonuclease Ⅲ (ExoⅢ).

## ExoIII in the commercial isothermal amplification diagnostic kits digested the ssDNA FQ reporter and probes

Due to the well-acknowledged dsDNA activities of ExoIII, most ExoIII-assisted diagnostic methods require ssDNA to form an appropriate dsDNA intermediate (*Table 2*). As a promising replacement for PCR technique (*Stringer et al., 2018*), RPA (*Piepenburg et al., 2006*; *Daher et al., 2016*), and MIRA (*Shen et al., 2019*; *Lu et al., 2022*) kits, as well as their derivative products, the MIRA- and RPA-ExoIII kits (*Zhi et al., 2022*; *Zhou et al., 2022*), have been widely used for nucleic acid detection. To explore if ExoIII contained in the commercial detection kits digests the ssDNA probe or primer, we used an FQ reporter and 5′ FAM-labeled ssDNA probes to test the commercial kits of MIRA-ExoIII and RPA-ExoIII, while the RPA and MIRA detection kits (without ExoIII) served as their controls. Our result showed that the RPA-ExoIII and MIRA-ExoIII detection kits exhibited significant fluorescence signals over time compared with those of the RPA and MIRA kits (*Figure 5A*). The distinct difference in fluorescence brightness between the ExoIII-presence group (MIRA-ExoIII and RPA-ExoIII) and ExoIII-absence group (MIRA and RPA) can be directly visualized under LED UV blue light (*Figure 5B*). These results suggested that the ssDNA FQ reporter was cleaved by ExoIII contained in the commercial detection kit, and this type of FQ reporter was unsuitable for developing the ExoIII-based detection.

Furthermore, to investigate the degradation degree of ssDNA caused by ExoIII in the detection kits, we tested a 5′ FAM-labeled ssDNA probe in these detection reactions. We found that 80% of the ssDNA probe added to the MIRA-ExoIII kits was digested to a 15-nt product (P1) (*Figure 5C and D*), while in RPA-ExoIII kit, 50% and 40% of the ssDNA probe were degraded into a~15 nt (P1) and ~5 nt (P2) in 20 min, respectively, presenting a more intensified degradation than MIRA-ExoIII (*Figure 5C and E*). It suggested ExoIII in the detection kits was capable of digesting the ssDNA probe or primer, leading to a low detection efficiency.

The different degrees of degradation between these two kinds of commercial kits might be caused by different amounts of T4 gp32 added, as T4 gp32 is a ssDNA-binding protein protecting the ssDNA from nuclease degradation (*Jensen and von Hippel, 1976*). To investigate whether the ssDNA-binding protein prevents ssDNA from ExoIII digestion, we incubated the ssDNA with two types of ssDNA-binding proteins (T4 gp32 or SSB) before ExoIII digestion. The ssDNA probe pre-incubated with 1 µl T4 gp32 (10 mg/ml), 1 µl, or 5 µl SSB (1.58 mg/ml) showed no digested band compared to the control (0 µl) (*Figure 5F, G and H*), which suggested the single-stranded DNA-binding protein was capable of protecting ssDNA from ExoIII digestion. These results indicated that adding more T4 gp32 could offset or minimize the effects of ExoIII′ ssDNase activity in the commercial kits. The findings also suggested that ssDNA-binding proteins may regulate in vivo the degradation of ssDNA mediated by ExoIII in DNA repair or metabolism processes.

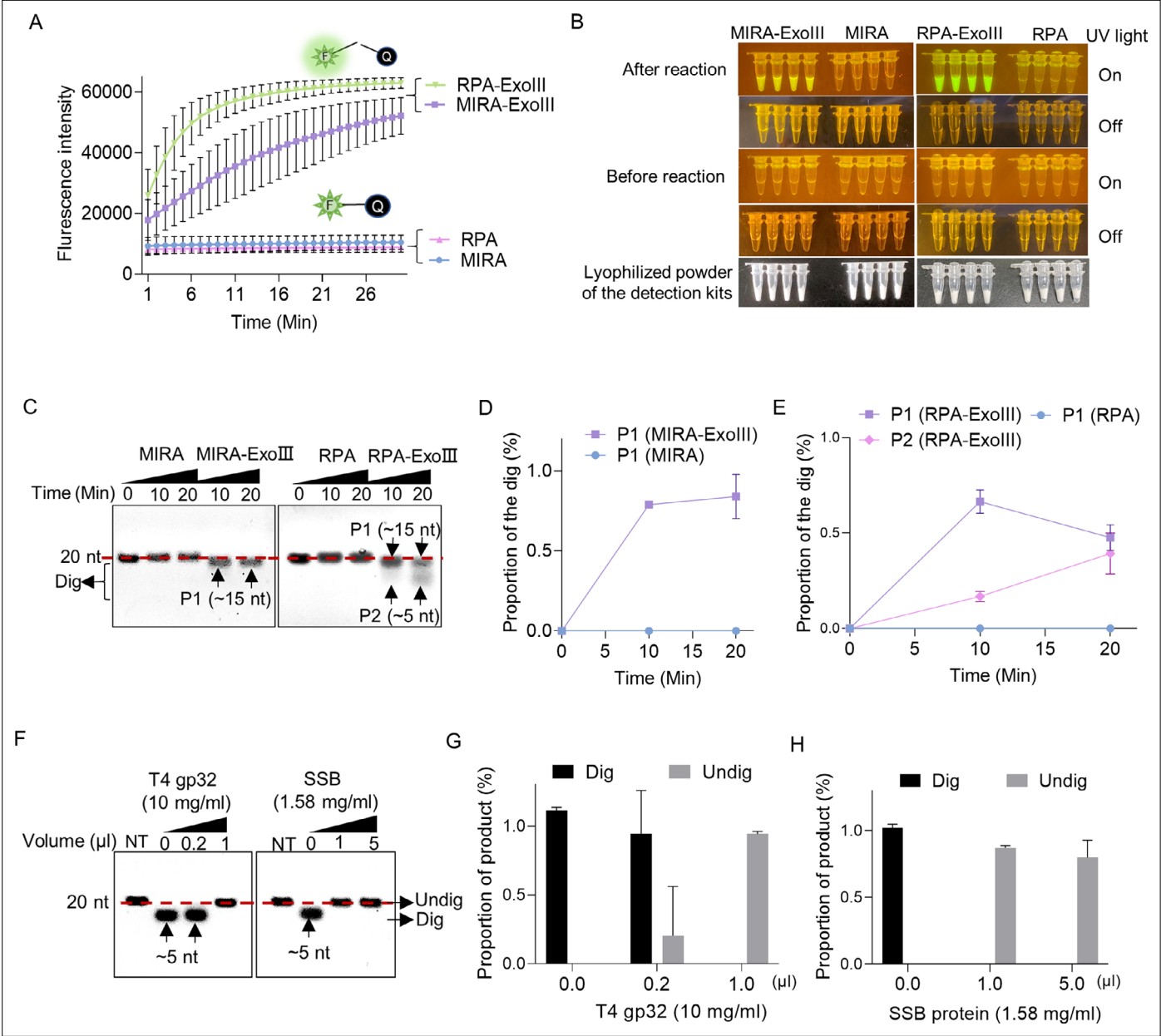

**Figure 5.** Exonuclease Ⅲ (ExoⅢ) in the isothermal amplification kits digested the ssDNA fluorescence-quenching (FQ) reporter and 5′ FAM-labeled ssDNA probes, while the single-stranded DNA binding protein protected the ssDNA probe from ExoⅢ digestion. (**A**) The fluorescence of the FQ reporter generated in four detection kits (MIRA, MIRA-ExoⅢ (containing ExoⅢ), RPA, and RPA-ExoⅢ (containing ExoⅢ)) was monitored for 30 min. MIRA and recombinase polymerase amplification (RPA) were the control of their corresponding counterparts, respectively. The average value of three repeats was calculated and plotted. (**B**) Before and after these four commercial kits were incubated with the FQ reporter for 30 min, the reaction tubes with fluorescence generated were visualized under LED blue light by the naked eye. MIRA-ExoⅢ indicates the ExoⅢ-containing MIRA kit; MIRA indicates the MIRA kit without ExoⅢ. (**C**) The 5′ FAM-labeled ssDNA probe (Probe 1) was incubated with the four detection kits for 20 min, and the reaction products were analyzed by gel electrophoresis. Undig, undigested; dig, digested. (**D, E**) The analysis on the gray intensities of the band was performed by ImageJ. The digested proportion was calculated by: (Intensity of the digested band)/(Intensity of the undigested band at 0 min). (**F**) Different amounts of single-stranded DNA binding protein (T4 gp32 or SSB (*E. coli*)) were incubated with the ssDNA oligo (Probe 1) for 10 min before being treated with ExoⅢ for 10 min. The reaction products of ExoⅢ digestion were separated by gel electrophoresis. The ssDNA oligo treated with deactivated ExoⅢ served as control. (**G, H**) The gray intensity of bands was measured by ImageJ, and the digested or undigested proportions were calculated by: (Intensity of the digested or undigested band)/(Intensity of the band produced in the treatment of deactivated ExoⅢ).

The online version of this article includes the following source data for figure 5:

**Source data 1.** Exonuclease Ⅲ (ExoⅢ) in the isothermal amplification kits digested the ssDNA fluorescence-quenching (FQ) reporter and 5′ FAM-labeled ssDNA probes.

## Key amino acid residues for ssDNase activity in ExoIII

To identify the key residues that determine the ssDNase activity of ExoIII, we predicted the structure of ExoIII-ssDNA by molecular docking. And the predicted residues (S217, R216, D214, W212, K176, R170, N153, K121, and Q112) (*Figure 6—figure supplement 1*) were expressed and purified for mutation analysis (*Figure 6A*, *Figure 6—figure supplement 2*). Residues (W212, F213, K121, Y109, and D151) reported to play essential roles in the exonuclease and AP-endonuclease activities of ExoIII (*Lee et al., 2022*; *Yoo et al., 2021*) were also mutated for analysis. The result of the fluorescence monitoring assay on the FQ reporter indicated the mutants of S217A, R216A, K176A, or R170A displayed equal cleavage efficiency with wild type ExoIII on the FQ reporter; D214A, W212A, F213A, Y109A, N153A, and D151N produced no fluorescence signal; K121A or Q112A showed much weaker cleavage than the wild type (*Figure 6B*). Meanwhile, gel electrophoresis on the digested products showed a consistent result with the fluorescence monitoring assay, but K121A exhibited undetectable digestion on the ssDNA probe (*Figure 6C and D*). These results indicated the residues (D214, W212, F213, N153, D151, K121, and Y109) were crucial in the catalyzing of ExoIII on ssDNA. As six of these critical residues (W212, F213, N153, Y109, K121, and D151) have also been found to play critical roles in the dsDNA exonuclease and AP-endonuclease activities of ExoIII (4), the ssDNase activity of ExoIII most likely shares the same active center with the other two enzymatic activities on dsDNA.

Among these mutants, K121A of ExoIII has been reported to have an AP-endonuclease activity on dsDNA (*Lee et al., 2022*). With the weakened ssDNase activity demonstrated in our study, K121A can replace the wild type to be applied in the AP-endonuclease-based commercial kits with minimized effects of ssDNase activity. By comparing the digested products generated by S217A, R216A, K176A, R170A, Q112A, and wild type, we found that R170A produced a larger fragment than other mutants and wild type (*Figure 6E*). It suggested that mutation of R170A affected the efficiency of ExoIII digestion on ssDNA. Based on the conservation indicator of all residues of ExoIII illustrated (https://consurfdb.tau.ac.il/index.php), these identified residues (W212, F213, D214, N153, Y109, K121, D151, and R170) were highly conserved among 300 homolog proteins (*Figure 6F and G*). The structure of ExoIII is constructed as a four-layered sandwich, and its active center on dsDNA is located at the valley between the two inner layers (*Lee et al., 2022*; *Mol et al., 1995*; *Figure 6G*).

## R170A has an attenuated AP-endonuclease activity

The residue of R170 is located with the protruding structure $\alpha_M$ helix of ExoIII, which helps ExoIII bind to dsDNA by stretching into the major groove (*Mol et al., 1995*). To investigate if the mutation of R170A also affects the AP-endonuclease and exonuclease activities of ExoIII on dsDNA, we designed specific dsDNA substrates for the two types of activities. On the AP-endonuclease substrate, the mutation of R170 took a much longer time (10 min) to digest compared to the wild type (1 min) (*Figure 6—figure supplement 3A and B*), suggesting the mutation seriously impaired the efficiency of AP-endonuclease activity. The digestion process is described in *Figure 6—figure supplement 3C*. On the exonuclease substrate, the mutant of R170A produced a much larger product than the wild type (*Figure 6—figure supplement 3D and E*). It might be because R170A retained the intact exonuclease activity on dsDNA but not on ssDNA, leading to a larger ssDNA intermediate compared to the wild type. The exonuclease activities of ExoIII on dsDNA and ssDNA involved in the digestion are described in *Figure 6—figure supplement 3F*. Altogether, we deemed that R170 in the $\alpha_M$ helix structure may help recognize or stabilize the AP site on dsDNA or ssDNA substrate for further docking into the active site, and the mutation of R170 affects the AP-endonuclease and ssDNase activities of ExoIII but still retains its dsDNA exonuclease activity.

## The proposed enzymatic action model of ExoIII on ssDNA

Since the ssDNase activity probably shares the same active center with its dsDNA-targeted activity, an appropriate ssDNA substrate for ExoIII should possess a 'V'-shaped micro-structure, similar to the ssDNA conformation in the B-formed dsDNA (*Figure 7A*; *Figure 7—figure supplement 1*). According to the ExoIII-ssDNA structure and the previously reported functions of the key residues (*Lee et al., 2022*), four primary functional units for enzymatic actions of ExoIII are defined: capturing unit (R170, at 3′ side of the cleavage site and three nucleotides away from it), phosphate-stabilizing unit (Y109, K121, and N153, at 5′ side of cleavage site and three nucleotides away from it), sugar ring-stacking unit (W212, F213, and D214) and phosphodiester-cleaving unit (D151), all of which also represent four

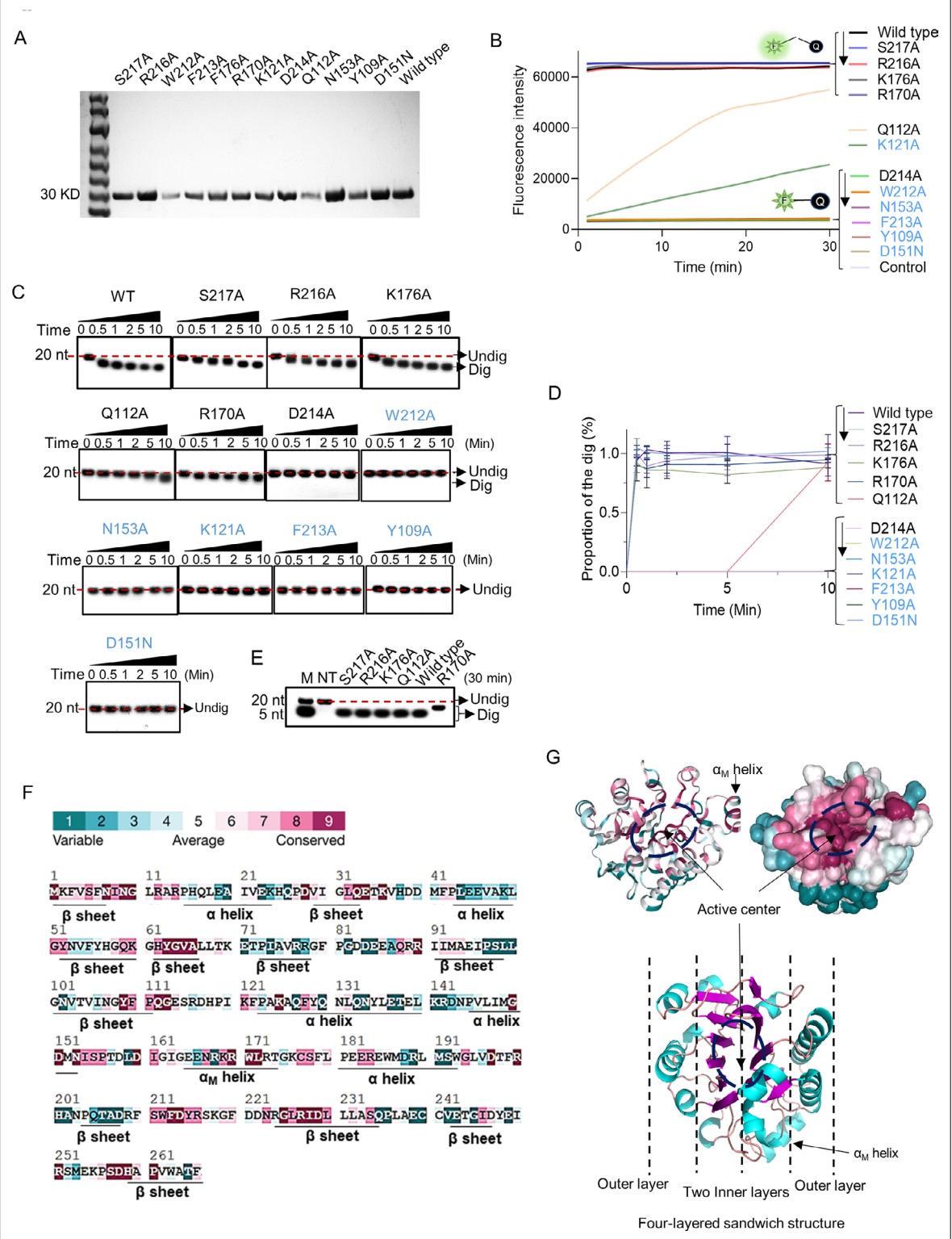

**Figure 6.** Point mutation of exonuclease III (ExoIII) identified the critical amino acid residues that determine the ssDNase activity. (**A**) All purified mutants and wild-type proteins used in the study were loaded into the 8% PAGE electrophoresis and stained with Coomassie Brilliant Blue. (**B**) The fluorescence-quenching (FQ) reporter (5 µM) was incubated with ExoIII mutants (2.5 µM) (S217A, R216A, D214A, F213A, W212A, K176A, D151N, R170A, N153A, K121A, Q112A, Y109A) and wild-type (WT) and the generated fluorescence was monitored for 30 min. FQ reporter treated with deactivated wild-type ExoIII was regarded as a control. The mutations in blue indicated they are previously-reported key residues for double-stranded DNA (dsDNA)-targeted activities of ExoIII. (**C**) The ssDNA oligo of Probe 3 (20 nt) was incubated with the ExoIII mutants, and the reaction products were

*Figure 6 continued on next page*

*Figure 6 continued*

analyzed by gel electrophoresis. (**D**) The gray intensity of the bands was determined by ImageJ. The average intensity value of three repeats was used to calculate the proportion of the digested band with: (Intensity of the digested band)/(Intensity of the undigested band at 0 min). (**E**) The digested products of S217A, R216A, K176A, R170A, Q112A, and wild type, after 30 min of incubation with ssDNA substrate, were compared with each other by gel electrophoresis. (**F**) The residue conservation of ExoIII (PDB ID:1AKO) was investigated using the ConSurf database (https://consurfdb.tau.ac.il/index.php). The conservation degree of ExoIII residues is calculated among 300 homologs, which the color scale indicates. (**G**) The carton and surface style of the ExoIII structure (PDB ID:1AKO) with conservation-indicated color is displayed (upper part). The four-layered sandwich structure of ExoIII is displayed by Pymol (lower part): the pink indicates β-sheet; the blue represents α-helices; the red indicates the random coils or turns.

The online version of this article includes the following source data and figure supplement(s) for figure 6:

**Source data 1.** The critical amino acid residues of exonuclease Ⅲ (ExoⅢ) for its ssDNase activity.

**Figure supplement 1.** The molecular docking of exonuclease Ⅲ (ExoIII) (PDB code: 1AKO) and ssDNA (NDB ID: 1S40) was performed at the HADDOCK docking server (https://wenmr.science.uu.nl/haddock2.4/), and the result was displayed in detail.

**Figure supplement 2.** All purified mutants and wild-type proteins used in the study were examined by the silver-stained SDS-PAGE gel.

**Figure supplement 2—source data 1.** The silver-stained SDS-PAGE gel.

**Figure supplement 3.** R170A mutant exhibited a weakened apurinic/apyrimidinic (AP)-endonuclease activity.

**Figure supplement 3—source data 1.** R170A mutant exhibited a weakened apurinic/apyrimidinic (AP)-endonuclease activity.

sequential steps in the digestion process (**Figure 7B**). Unit 1(R170), located at the surface of ExoIII and functioning at the 3′ side of the cleavage site, is most likely the first contact with the substrate, the involvement of which naturally drives the endonuclease action on ssDNA. In the endonuclease model, interactions of unit 1 and unit 2 with ssDNA substrates can help convert the ~6 nucleotides at 3′ end of ssDNA to form a 'V' shape intermediate, suitable for the following actions of unit 3 and unit 4. Certain ssDNA substrates, such as the consecutive identical bases (**Figure 4**), might have difficulty in converting to the necessary conformation, resulting in the inhibition of ExoIII digestion; Meanwhile, the substrates with 'V' conformation internally possibly attract units 2 and 3 of ExoIII binding to the site and trigger the endonucleolytic digestion, which explains the products of endonuclease activity in the mass spectrometry analysis (**Figure 3**). In the exonuclease model, some types of ssDNA (possibly short ssDNA or others with suitable micro-structure) might directly enter the active center without contacting unit 1 and sequentially be processed by units 2, 3, and 4 (**Figure 7C**).

Combining with the structure analysis and the digestion results of **Figures 2 and 3**, we proposed the enzymatic actions of ExoIII digestion on ssDNA: the ssDNA substrate tends to be captured by the outer-layered unit 1, then stabilized by the inter-layered units 2 and 3, finally cleaved by unit 4 of ExoIII, producing short fragments of ssDNA (endonuclease activity). The short ssDNA (~3 nt) or others with suitable micro-structure might directly enter into the active center and be processed by unit 2, unit 3, and unit 4 into nucleotides (exonuclease activity) (**Figure 7D**). The model reflects a distributive catalyzation of ExoIII on ssDNA. In summary, ssDNA substrate is likely to undergo fragmentation by the endonuclease activity of ExoIII, and the resulting small fragments are subsequently digested by the exonuclease activity of ExoIII into mononucleotides. Without the base-pairing of complementary strands, the conformation of ssDNA is easily affected by ion concentration, interacted protein, temperature, and other factors in the solution. All these elements together forge the specific catalytic pattern of ExoIII on ssDNA.

## ExoIII digested the dsDNA structures containing 3′ end ssDNA

To explore the biological significance of the ssDNase activity of ExoIII, we examined several dsDNA substrates containing ssDNA (dsDNA with 3′ssDNA flap, dsDNA with 3′protruding ssDNA tail, bubble, and fork-shaped dsDNA) that might occur in the biological process. After being treated with ExoIII, our result demonstrated that ExoIII was capable of digesting the dsDNA with a 3′ flap (**Figure 8A**). The 3′ flap-removing capability of ExoIII may originate from its ssDNase activity (**Figure 8B**). The analysis of the gray intensity indicated the dsDNA with 3′ flap was digested with time (**Figure 8C**). Based on our results and the gap creation ability of ExoIII previously reported (**Yoo et al., 2021**), we proposed the novel biological role of ExoIII in DNA repair: when the 3′ flap occurs, ExoIII recognize and remove the ssDNA flap by the ssDNase activity; then it continues to digest the ssDNA on dsDNA and create a gap; the resulting intermediate recruits the DNA polymerase to re-synthesize the complementary strand and fill the ssDNA gap; finally, the nick left is sealed by DNA ligase (**Figure 8D**).

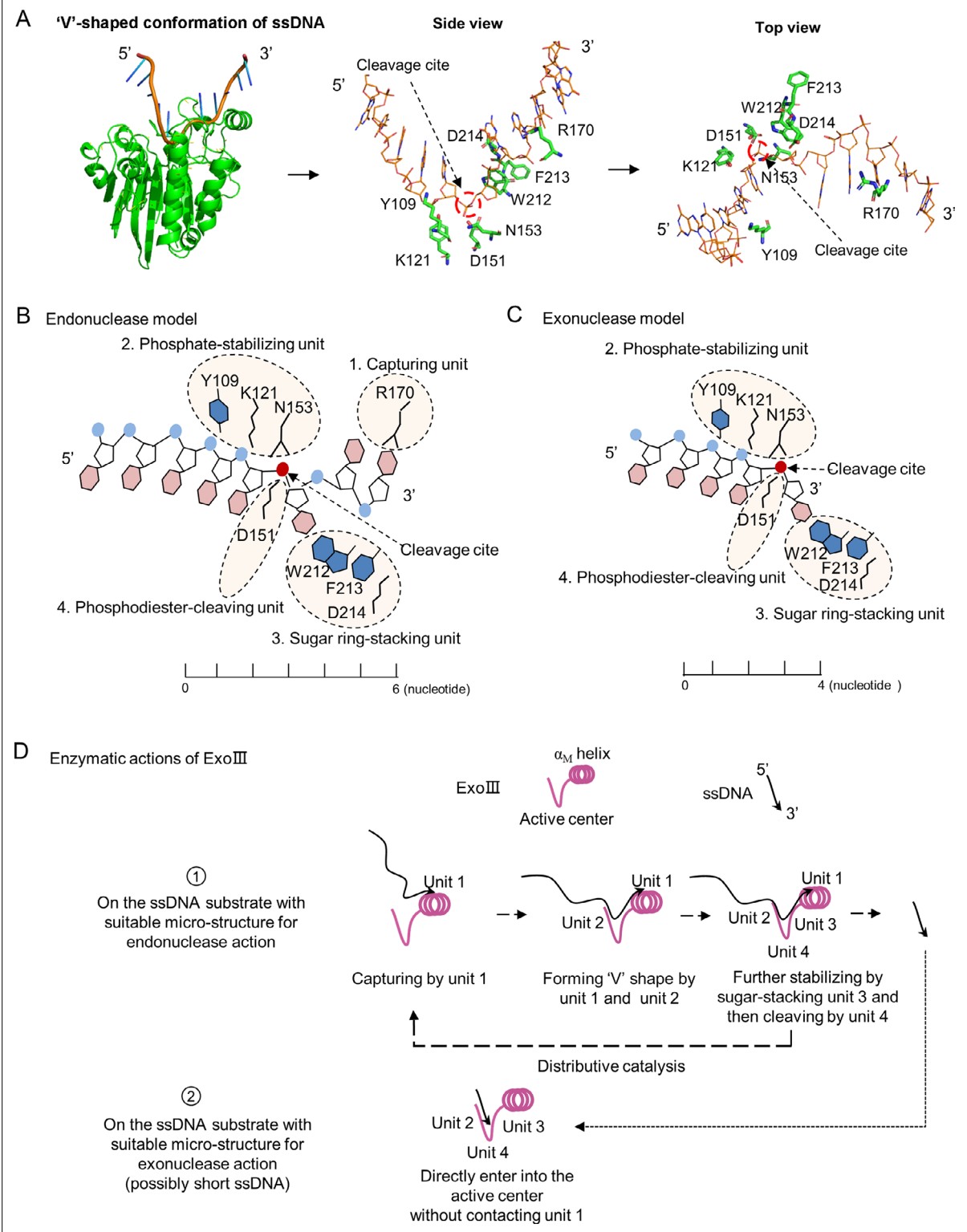

**Figure 7.** A theoretical model for the enzymatic actions of exonuclease Ⅲ (ExoIII) on ssDNA is proposed. (**A**) The ExoIII-ssDNA structure was obtained by structure alignment between ExoIII (PDB:1AKO) and APE1-dsDNA (PDB: 5WN5) by Pymol. The key residues and the cleavage site of ExoIII are labeled in situ and presented by side and top view. (**B**) According to the ExoIII-ssDNA structure and the previously reported functions of the key residues (4), four functional units of ExoIII on ssDNA are primarily defined: capturing unit (R170, at 3′ side of the cleavage site and three nucleotides away from it), phosphate-stabilizing unit (Y109, K121, and N153, at 5′ side of cleavage site and three nucleotides away from it) and sugar ring-stacking unit (W212, F213, and D214) and phosphodiester-cleaving unit (D151). In the model of endonuclease, Unit 1(R170), located at the surface of ExoIII, is most likely the

*Figure 7 continued on next page*

*Figure 7 continued*

first contact with the substrate, and binding at the 3′ side of the cleavage site naturally drives the endonucleolytic cleavage on ssDNA. Interactions of unit 1 and unit 2 with substrates may help convert the ~6 nucleotides at 3′ end of ssDNA to form a 'V' shape intermediate for docking into the active center and being processed by unit 3 and unit 4. (**C**) In the exonuclease model, some types of ssDNA (possibly short ssDNA or others with suitable micro-structure) directly enter into the active center without contacting unit 1, and sequentially processed by unit 2, 3 and 4. (**D**) The ssDNA substrate tends to be captured by the unit 1, then stabilized by the unit 2 and 3, and finally cleaved by unit 4 of ExoIII, producing short fragments of ssDNA (endonuclease activity), which may reflect the distributive catalysis of ExoIII on ssDNA. The short ssDNA (~3 nt) or others with suitable microstructure would directly enter into the active center and be processed by units 2, 3, and 4 into mononucleotides (exonuclease activity). Most ssDNA substrates might be digested by both endonuclease and exonuclease activity of ExoIII.

The online version of this article includes the following figure supplement(s) for figure 7:

**Figure supplement 1.** Structure alignment was performed between exonuclease III (ExoIII) (PDB ID: 1AKO) and APE1-dsDNA (PDB ID: 1DE8 and NDB ID: 5WN5) complex through TM-align (Version 20190822) (https://zhanggroup.org/TM-align/).

We next tested its ability to cleave the dsDNA substrates with a protruding ssDNA (≥4 nt) at the 3′ end, previously reported as resistant to ExoIII digestion (*Henikoff, 1984*). Our results on the dsDNA with a 10-base protruding structure indicated that ExoIII could rapidly digest the dsDNA with a 3′ end 10-base ssDNA tail (*Figure 8—figure supplement 1A and B*). But for the dsDNA with 4-base protruding ssDNA at the 3′ end, the digestion occurred at 10 min, suggesting the efficiency of digestion significantly decreased to about one-tenth of the 10-base protruding structure (*Figure 8—figure supplement 1C and D*). Unlike the previous study (*Henikoff, 1984*), our results demonstrated that ExoIII could catalyze dsDNA structure with 3′ end protruding bases (≥4 nt). Furthermore, we found when the 3′ end 4-base protruding ssDNA was replaced with four adenylates, the dsDNA showed no digestion over 10 min (*Figure 8—figure supplement 1E and F*), suggesting the 3′ A$_4$-protruding ssDNA was resistant to the ssDNase activity of ExoIII and protected the dsDNA duplex from ExoIII digestion. Therefore, dsDNA with 3′ A$_4$-protruding ssDNA can be used for related enzymatic research of ExoIII or as the aptamer of biosensors, as the previous typical substrate may not protect itself from being cleaved by the ssDNase activity and the following exonuclease of ExoIII. Based on the structural analysis that indicated the α$_M$ helix was exactly 4~5 nt apart from the AP site (located in the active center) (*Figure 8—figure supplement 1G and H*), we speculated that the lower efficiency of digestion on the four-base protruding dsDNA might be because the topology of dsDNA-ssDNA junction site was not well recognized by α$_M$ helix, which was unfavourable for the ssDNA further docking into the active center.

The fork structure was slightly digested by ExoIII in the first minute and completely degraded over 10 min (*Figure 8—figure supplement 2A*). To test if the ssDNA binding protein affects the digestion of ExoIII, the fork structure was incubated with T4 gp32 for 10 min before adding ExoIII. The result indicated that 1 µl T4 gp32 (10 mg/mL) protected over 40% of the substrate from digestion, significantly higher than 0.2 µl (~10%) (*Figure 8—figure supplement 2A*). This protection effect suggested that ExoIII started the digestion of the structure from the 3′ end ssDNA, and the binding of T4 gp32 to the 3′ end ssDNA prevented the dsDNA from being digested (*Figure 8—figure supplement 2B*). The analysis on gray intensity showed ExoIII was able to digest the fork dsDNA structure and the single binding protein delayed the process by binding to the ssDNA part (*Figure 8—figure supplement 2C and D*). The bubble structure showed no apparent digestion over time (*Figure 8—figure supplement 2E and F*). Collectively, these results indicated ExoIII may function in more types of dsDNA structures with 3′ ssDNA in DNA-repair or -metabolism processes.

## Discussion

Defining the ssDNase activity of ExoIII is crucial for understanding its biological roles and guiding the current diagnostic applications. In this study, we adopted the sensitive ssDNase indicator FQ reporters, a range of short ssDNA substrates (~20 nt), and mass spectrometry analysis for prudent confirmation and systematic characterization of the ssDNase activity of ExoIII. Our results uncovered that ExoIII possessed highly efficient 3′→5′ exonuclease and endonuclease activities on ssDNA, with the cleavage site at the 5′ side bond of the phosphodiester. As the discovery may remould the roles of ExoIII in its industrialized application and biological understanding in the DNA repair process, we assessed its effects by testing the popular commercialized kits containing ExoIII and investigated its

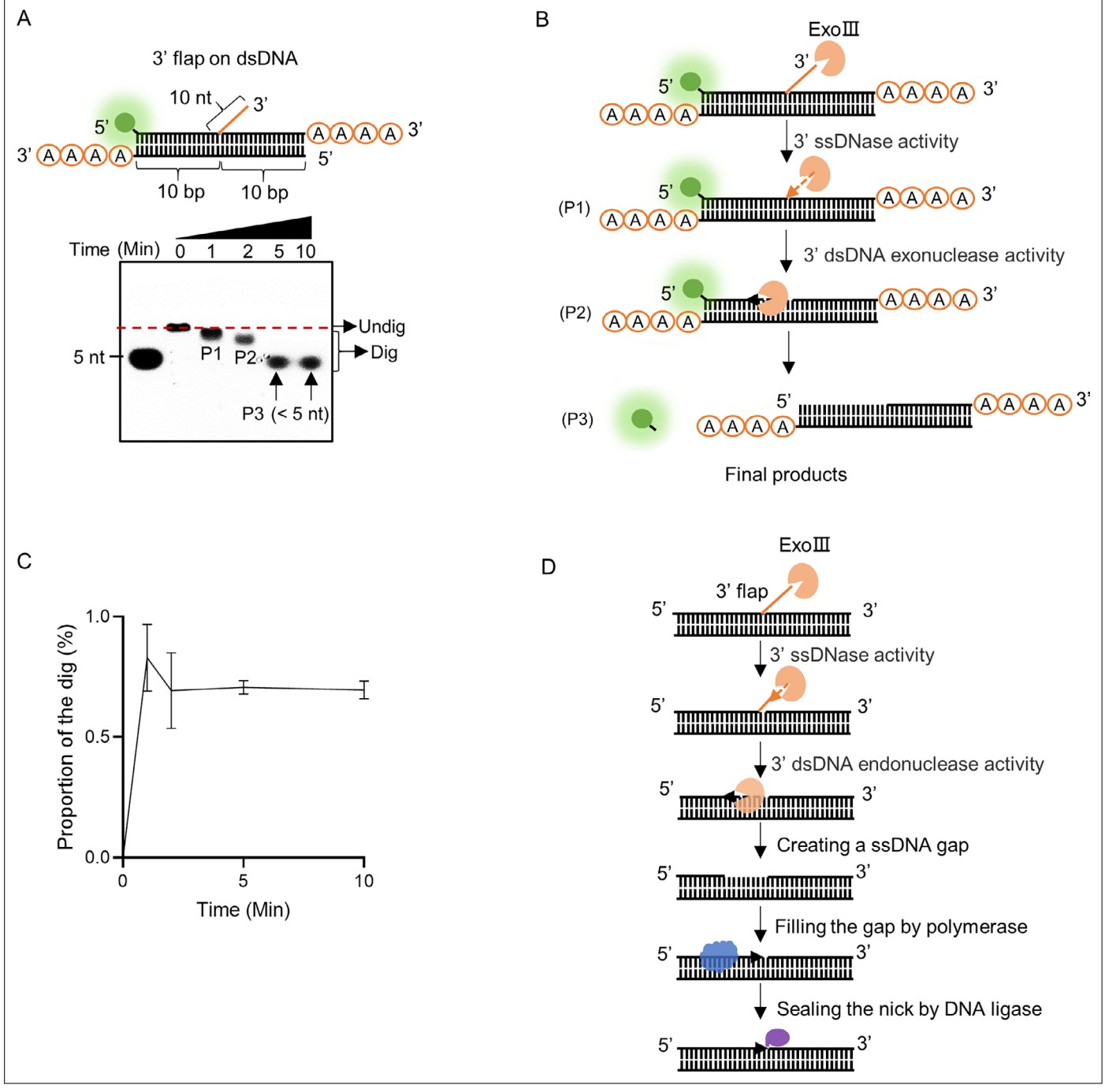

**Figure 8.** Exonuclease Ⅲ (ExoⅢ) efficiently digested the 3' flap ssDNA on double-stranded DNA (dsDNA). (**A**) The constitution of the dsDNA structure with 3' flap is displayed. The time course analysis of ExoⅢ (2.5 µM) digestion on the structure (5 µM) was performed, and the products were analyzed by gel electrophoresis. (**B**) According to the activities of ExoⅢ on ssDNA and dsDNA, the process of enzymatic reaction on the dsDNA structure was outlined. (**C**) The digested proportions of ssDNA substrates were calculated by: (gray intensity of the digested band)/(gray intensity of the undigested band at 0 min) and plotted based on an average value of three repeats. (**D**) Based on our result and the gap-creation ability of ExoⅢ (*Yoo et al., 2021*), a novel biological role of ExoⅢ in DNA repair was proposed: when the 3' ssDNA flap occurs on dsDNA during biological processes such as base excision repair (BER) or DNA replication, ExoⅢ recognizes and removes the ssDNA flap by its ssDNase activity; then it continues to digest and create a ssDNA gap on the dsDNA by exonuclease activity; the resulted intermediate recruits the DNA polymerase to re-synthesize the complementary strand; finally, the nick left is sealed by DNA ligase.

The online version of this article includes the following source data and figure supplement(s) for figure 8:

**Source data 1.** Exonuclease Ⅲ (ExoⅢ) efficiently digested the 3' flap ssDNA on double-stranded DNA (dsDNA).

*Figure 8 continued on next page*

*Figure 8 continued*
**Figure supplement 1.** The double-stranded DNA (dsDNA) substrates with 3′ protruding bases (≥4 nt) were digested by exonuclease Ⅲ (ExoⅢ).
**Figure supplement 1—source data 1.** Exonuclease Ⅲ (ExoⅢ) digested the double-stranded DNA (dsDNA) substrates with 3′ protruding bases.
**Figure supplement 2.** Exonuclease Ⅲ (ExoⅢ) digested the fork double-stranded DNA (dsDNA) structure with 3′ end ssDNA, while the ssDNA binding protein T4 gp32 suppressed the digestion.
**Figure supplement 2—source data 1.** Exonuclease Ⅲ (ExoⅢ) digested the fork double-stranded DNA (dsDNA) structure with 3′ end ssDNA.

possible biological functions through the dsDNA structures containing ssDNA that might occur in vivo. Our results showed that ExoIII digested ssDNA probes or primer in the isothermal amplification kit and dsDNA structures containing 3′-end ssDNA. To minimize the effects of ExoIII′ ssDNase activity, we advise: (a) adding more single-strand DNA binding protein T4 gp32 protects the primer or ssDNA probe from degradation of ExoIII in the isothermal amplification kits; (b) adding four or more consecutive adenylates at the 3′-end of the dsDNA structure or ssDNA helps inhibit the degradation of ExoIII; (c) with limited ssDNase activity, K121A and R170A mutants can replace the wild type and be applied in the AP-endonuclease and exnuclease based diagnostics, respectively; (d) raise the reaction temperature to ≥42°C, as the ssDNase activity of ExoIII was found highly suppressed in the temperature range. Considering that the ssDNase activity of ExoIII shares the same active center as its dsDNA-targeted activity, examining its biological roles in vivo by individual point mutation seems challenging. But digesting the dsDNA with 3′ flap ssDNA or 3′ protruding ssDNA tail in vitro indicated the ssDNase activity of ExoIII may confer a 3′ flap-cleaving ability, suggesting it might be competent to deal with more types of DNA lesions. During this, the ssDNA-binding proteins may play a regulatory role in ExoIII-related ssDNA degradation in vivo. As a multi-functional nuclease, the enzymatic activities of ExoIII identified so far are summarized in *Table 3*.

The endonuclease activity of ExoIII on ssDNA was confirmed for the first time by mass spectrometry analysis on Probe 1 and Probe 2. It might explain why digested products of ExoIII and LbCas12a only accounted for 50~80% of their deactivated counterparts (*Figure 3C, G and K*). The lost 20~50% fluorescence is possibly caused by ExoIII or LbCas12a cleaving the substrates and forming products with different lengths, which travelled in the gel at different speeds, finally leading to being dispersed or invisible under UV light. Combining these results with the analysis of the constructed crystal structure of ExoIII-ssDNA, we proposed a theoretical model for understanding the underlying mechanism of endonuclease and exonuclease actions of ExoIII on ssDNA. Steven A. Benner and his coworkers observed that ExoIII could bypass two consecutive nucleotides with phosphorothioate modification and continued deleting the ssDNA (*Yang et al., 2007*), which might be well-explained by endonuclease activity in the model. In addition, its human homolog APE1 has been reported to have a strong binding affinity to the G-rich oligo (5′-AGGGCGGTGTGGGAAGAGGGAAGAGGGGGAGG-3′, a 32-nt sequence of the oncogene KRAS promoter) (*Pramanik et al., 2022*). Thus, ExoIII also possibly bound to the G-rich region of Probe 2 and activated its endonuclease activity, producing more middle fragments than the other two ssDNA probes, as indicated in mass spectrometry analysis.

Another interesting finding is that the consecutive identical bases of ssDNA stalled the digestion of ExoIII. In the early 1960s, the inactivity of ExoIII on ssDNA was coined by Kornberg's laboratory based

**Table 3.** An updated list for ExoIII activities.

| Substrates | Preferred structures | Activities | Products | Key references |
|---|---|---|---|---|
| | With 3′ phosphonate | 3′ phosphatase | Phosphate | *Richardson and Kornberg, 1964* |
| | Containing AP | Endonuclease | ssDNA gap on DNA | *Rogers and Weiss, 1980* |
| dsDNA | Natural DNA | Exonuclease | Phosphate, 5′ mononucleotides, ssDNA | *Richardson et al., 1964* |
| RNA | RNA/DNA hybrid | RNase H | Mono-ribonucleotide, ssDNA | *Weiss et al., 1978* |
| | Containing AP | Active (less than dsDNA) | Oligos | *Shida et al., 1996* |
| ssDNA | Nonstructural ssDNA | Exonuclease and endonuclease activities | Mononucleotide, Oligos | This study |

on an ssDNA oligo of five consecutive Ts (5′ pTpTpTpTpT 3′) (*Richardson et al., 1964*) (which might not be sufficiently prudent to conclude that ExoIII is inactive on all ssDNA). In 2007, Steven A. Benner and his coworkers pointed out for the first time that ExoIII could degrade ssDNA by incubating with two long ssDNA oligos (51 nt) (*Yang et al., 2007*). An approximate 43-mer band occurred as the superior product when the concentration of ExoIII decreased, but the reason remains unclear. Here, based on our conclusion, the 43-mer intermediate exactly corresponds to the product generated when ExoIII stalled its digestion ahead of the consecutive GGG (Position 40–42) of the ssDNA substrate in 3′→5′ direction (5′-GCGTAATACGACTCACTATAGACGAGCGTACTTTAGTGA**GGG**TTAATTCGC-3′). In addition, adding 10T bases to an ssDNA at the 3′ end has been indicated to suppress ExoIII digestion (*Cai et al., 2014*); a *Saccharomyces cerevisiae* homolog of ExoIII, CCR4 (an ssDNA exonuclease), has also been reported to have a slower degradation on the substrate with 3′ end five consecutive As and was unable to cleave when the ssDNA contained ten consecutive As within (*Chen et al., 2002*). Based on our proposed model, the sequence feature of consecutive identical bases in ssDNA may create an unfavorable local microstructure that can impede ExoIII digestion.

Our work uncovers the long-underestimated ssDNase activity of ExoIII, dissects the underlying enzymatic behaviors of ExoIII on ssDNA and advances the mechanism understanding of its biological roles in DNA repair. Future investigations might be conducted: (1) crystal structure of the ExoIII-ssDNA complex (endonuclease and exonuclease model); (2) potential biological roles of the ssDNase activity in vivo. A better understanding of the underlying enzymatic actions of this multifunctional enzyme will facilitate efficient and reliable applications of ExoIII-based nucleic acid diagnostics.

# Materials and methods

### Key resources table

| Reagent type (species) or resource | Designation | Source or reference | Identifiers | Additional information |
|---|---|---|---|---|
| Gene (*E. coli*) | ExoⅢ | GenBank | GeneID:946254 | |
| Strain, strain background | Rosetta(DE3) Competent Cells | LGC Biosearch Technologies | 70954 | |
| Peptide, recombinant protein | ExoⅢ | NEB | NEB#M0206V | |
| Peptide, recombinant protein | T7 exonuclease | NEB | NEB#M0263L | |
| Peptide, recombinant protein | LbCas12a | NEB | NEB#M0653S | |
| Peptide, recombinant protein | T4 gp32 | Beyotime Biotech. Inc, China | D7057S | |
| Peptide, recombinant protein | SSB (*E. coli*) | Beyotime Biotech. Inc, China | P7415 | |
| Recombinant DNA reagent | pET-30a (plasmid) | Addgene | EMD Biosciences | Protein expression |
| Commercial assay or kit | HisSep Ni-NTA 6FF Chromatography | Yeasen Biotechnology Co., Ltd, China | 20503ES10 | Protein purification |
| Commercial assay or kit | ClonExpress II One Step Cloning Kit | Vazyme Biotech, China | C112 | |
| Commercial assay or kit | T7 in vitro transcription kit | Vazyme Biotech, China | TR101 V22.2 | crRNA synthesis |
| Commercial assay or kit | TURBO DNase | Thermo Fisher Scientific, Massachusetts, USA | AM2238 | crRNA synthesis |
| Commercial assay or kit | RPA | TwistAmp Basic | TABAS03KIT | |
| Commercial assay or kit | MIRA (Basic type) | Amplification Future company | WLB8201KIT | |
| Commercial assay or kit | RPA | TwistAmp exo | TAEXO02KIT | |

*Continued on next page*

*Continued*

| Reagent type (species) or resource | Designation | Source or reference | Identifiers | Additional information |
|---|---|---|---|---|
| Commercial assay or kit | MIRA (Fluorescence type) | Amplification Future company | WLE8202KIT | |
| Sequence-based reagent | FQ reporter | This paper | Sunya Biotechnology Co., China | 5'-FAM-TTATT-BHQ1-3' |
| Sequence-based reagent | Probe 1 | This paper | Sunya Biotechnology Co., China | CAAACCCAGAGCCAATCTTATCT |
| Sequence-based reagent | Probe 2 | This paper | Sunya Biotechnology Co., China | CGGGTGGGCGGAAAACTATTTC |
| Sequence-based reagent | Probe 3 | This paper | Sunya Biotechnology Co., China | AGTCCGTTTGTTCTTGTGGC |
| Sequence-based reagent | crRNA | This paper | | GGUAAUUUCUACUAAGUGUA GAUAACAGCACAUGCAGAAUCAU |
| Software | PyMOL | DeLano Scientific | | |
| Software | Mx3500P | Agilent | | Fluorescence monitoring |
| Software | Graphpad | Dotmatics | Version 9.0.0 | |

## Material and reagents

To avoid any potential secondary structure formed, we used short ssDNA oligos (5~23 nt) (Sunya Biotechnology Co., Hangzhou, Zhejiang Province, China) as the substrate for the nuclease treatment. The FQ reporter was a synthesized ssDNA oligo labeled with a fluorophore (FAM or FITC) at its 5' end and a quencher (BHQ1) at its 3' end. The FQ ssDNA reporter is a sensitive sensor of ssDNA-cleaving activity, enabling a fluorescence signal's visible emission upon ssDNase-like activity-mediated cleavage (*Kaminski et al., 2021*). The other reporter (termed the FB reporter) was an ssDNA oligo labeled with FAM at the 5' end and biotin at the 3' end, which is tailored for use in lateral flow strips and yields a positive result upon cleavage by ssDNase activity, as indicated by the appearance of two lines on the strip (*Kaminski et al., 2021*). All the sequences and related modifications are detailed in *Supplementary file 2*. ExoIII (NEB#M0206V), APE1 (NEB#M0282S), T7 exonuclease (T7 exo) (NEB#M0263L) and LbCas12a (NEB#M0653S) were bought from New England Biolabs (Ipswich, MA, USA). The solutions of $MgCl_2$ and EDTA-2Na (referred to as EDTA) (Sinopharm Chemical Reagent Co., Ltd. Shanghai, China) were respectively prepared with deionized water to concentrations of 10 mM (pH = 7) and 100 mM (pH = 8). The 5' FAM-labeled ssDNA constructed with 5 nt, 15 nt, 20 nt, and 30 nt adenylates were synthesized and used as the ssDNA marker in the gel electrophoresis.

## Protein expression and purification

The protein expression and purification followed the method in the previous study (*Lee et al., 2022*). The wild type of ExoIII gene (*E. coli*) (GeneID:946254) and the DNA fragments of mutants (S217A, R216A, D214A, F213A, W212A, K176A, D151N, R170A, N153A, K121A, Q112A, Y109A) were cloned into pET-30a (6×His–ExoIII) by using ClonExpress II One Step Cloning Kit (Vazyme Biotech, Nanjing, China). These vectors were transformed into Rosetta (DE3) *E. coli* (LGC Biosearch Technologies, Hoddesdon, UK). After the plasmids were confirmed by sequencing, cells were placed in a 37 °C incubator (adding 100 µg/mL kanamycin) and shaken until the OD600=0.6. Then, the culture was added with 1 mM IPTG (final concentration) and incubated at 16 °C shaking incubator for 16 hr. Cells were collected by centrifugation at 7500×g for 10 min and resuspended in 20 mL PBS buffer (137 mM NaCl, 2.7 mM KCl, 10 mM $Na_2HPO_4$, and 1.8 mM $KH_2PO_4$, pH = 7.8). The suspensions were sonicated with 4 s pulse-on and 6 s pulse-off (*P*=150 W) for 15~30 min. Products were centrifuged at 15000×g for 15 min, and then the supernatants were retained and filtered by micro syringe filters (0.45 µm). The protein was purified with HisSep Ni-NTA 6FF Chromatography Column (Yeasen Biotechnology Co., Ltd., Shanghai, China). After dialysis in PBS buffer, these proteins were aliquoted with 20% glycerol

and stored at –80 °C. The tag does not affect the enzymatic activity of ExoIII (*Lee et al., 2022*; *Figure 6—figure supplement 3*).

## Fluorescence monitoring assay on FQ reporter

In the assay, 5 µM ssDNA FQ reporter was incubated with ExoIII (5 U/µl), T7 exo (5 U/µl), APE1 (5 U/µl) or LbCas12a-crRNA-activator complex (0.1 µM). To form the LbCas12a-crRNA-activator complex, LbCas12a crRNA, and the activator dsDNA are pre-incubated at 37 °C for 10 min before incubating with FQ reporter. The corresponding buffer for each nuclease was added based on the manufacturer's document, and the total reaction volume was supplemented to 10 µl by the nuclease-free water. During the incubation, the fluorescence value was recorded every minute for over 30~60 min at 37 °C by Mx3500P (Agilent, CA, USA). The endpoint detection here was defined as detecting the accumulated fluorescence once at the end of the reaction. The image of fluorescence excited by LED blue light was captured in an Ultra Slim LED illuminator (Miulab, Hangzhou, Zhejiang Province, China).

## Enzymatic reaction on ssDNA

The enzymatic reaction contained 5 µM fluorophore-labeled ssDNA oligos, 5 U/µl commercial ExoIII or 2.5 µM purified wild-type and mutant ExoIII, 1 µl 10 x NEbuffer #1. The total reaction volume was supplemented to 10 µl by nuclease-free water. Before being treated with T7 exonuclease (T7 exo) (5 U/µl), APE1 (5 U/µl), ExoIII or LbCas12a-crRNA-activator complex (0.1 µM), the ssDNA oligos were incubated at 95 °C for 5 min, then cooled down in the ice. The nucleases were deactivated by heat at 95 °C for 5 min and used as controls (D-T7 exo, D-APE1, D-ExoIII, and D-LbCas12a). After incubating in a 37 °C water bath for 40 min, enzymatic reaction products were subjected to gel electrophoresis.

Different dsDNA substrates were prepared to investigate the roles of ExoIII in DNA repair. An equivalent amount of complementary ssDNAs (10 µM) was mixed and annealed to each other by heating at 95 °C for 5 min and cooling down at room temperature for 30 min. The generated dsDNA (5 µM) was incubated with ExoIII (2.5 µM), and the reaction product was then analyzed by gel electrophoresis.

## The trans-cleavage reaction of LbCas12a on ssDNA

For the reaction of LbCas12a on ssDNA, crRNA and dsDNA activators for the trans-cleavage activity of LbCas12a were prepared. The crRNA was synthesized using the T7 in vitro transcription kit (Vazyme Biotech, Nanjing, China). The dsDNA template for transcription was designed by annealing a single-stranded DNA (ssDNA) comprising T7 promotor and crRNA sequence (SUNYA Biotechnology Co., Ltd, Zhejiang, China) to its complementary oligonucleotides. A total volume of 20 µl was prepared by mixing template dsDNA (~100 ng), 2 µl of T7 RNA polymerase, 8 µl of NTP buffer, 2 µl of reaction buffer, and nuclease-free water. The reaction mix was incubated at 37 °C water bath for 16 hr. The transcription product was treated with TURBO DNase (Thermo Fisher Scientific, Massachusetts, USA) at 37 °C for 30 min and purified by Monarch RNA Cleanup Kit (New England Biolabs, Massachusetts, USA). The concentration of crRNA was determined by Nanodrop 2000 (Thermo Fisher Scientific, Massachusetts, USA). The dsDNA activator for the trans-cleavage activity of LbCas12a was created by annealing complementary ssDNAs containing PAM and the target sequence against crRNA. To form the LbCas12a-crRNA-activator complex, LbCas12a crRNA, and the activator dsDNA are pre-incubated at 37 °C for 10 min before incubating with an ssDNA probe or FQ reporter. The sequences are presented in *Supplementary file 2*.

## The fluorescence-based gel electrophoresis assay

The gel was prepared with a concentration of 6% (wt/vol) agarose (HydraGene Co., Ltd, Xiamen, Fujian Province, China). After incubating with ExoIII, the reaction mixtures mixed with a 6x loading buffer (Takara, Beijing, China) were loaded into the gel. The electrophoresis (EPS300, Tanon, Shanghai, China) was carried out at 130 V for ~40 min in Tris-acetate-EDTA (TAE) running buffer (400 mM Tris, 25 mM EDTA, adding acetic acid to pH = 7.8). A dark environment was required during electrophoresis to slow down fluorescence extinction. The abundance of reaction products was indicated by fluorescence bands under UV light, which was visualized and imaged by a multifunctional ultra-sensitive imaging system (Shenhua Science Technology Co. Ltd., Hangzhou, Zhejiang Province, China).

## Mass spectrometry analysis

After 5 µM fluorophore-labeled ssDNA oligo was incubated with 5 U/µl ExoIII at 37 °C for 0.5 h, the enzymatic reaction was stopped by heating at 95 °C for 5 min. The reaction product was then analyzed

by mass spectrometry. The reaction containing only ssDNA oligo or ExoIII was prepared as the control. Before incubation at 37 °C, the ssDNA oligos were pre-incubated at 95 °C for 5 min and cooled in the ice. Hippo Biotechnology Co., Ltd. performed the mass spectrometry (Thermo LTQ-XL, Massachusetts, USA) analysis on the samples (Huzhou, Zhejiang, China). The detection range of molecular weight in mass spectrometry is 500~20,000 Da. Electron Spray Ionization (ESI) was used to ionize the molecules.

## Molecular docking

Molecular docking of ExoIII (PDB code: 1AKO) and an ssDNA (NDB ID: 1S40) was performed in a webserver (https://wenmr.science.uu.nl/haddock2.4/; *Honorato et al., 2021*; *van Zundert et al., 2016*; *de Vries et al., 2010*). Before the docking, we performed structure alignment or superimposing on ExoIII and APE1-dsDNA (PDB ID: 1DE8 and NDB ID: 5WN5) complex through TM-align (Version 20190822) (https://zhanggroup.org/TM-align/; *Zhang and Skolnick, 2005*). We extracted the structure of the ExoIII-ssDNA complex by PyMOL (*Delano, 2002*) from the superimposing. Then residues of ExoIII located within 5 Å of the ssDNA and the last four nucleotides at the 3' end of the ssDNA were selected to input in the HADDOCK website for molecular docking. Docking parameters were set by default. The output structure of ExoIII-ssDNA with the highest score was visualized by PyMOL for further analysis.

## Evaluation of the ExoIII' ssDNase activity in the commercial diagnostic kits

To examine the effects of ExoIII' ssDNase activity on the commercial isothermal amplification detection kit containing ExoIII, we tested four kits including RPA (recombinase polymerase amplification) with ExoIII (TwistAmp exo) or without ExoIII (TwistAmp Basic) (Abbott, Illinois, US), MIRA (Multienzyme Isothermal Rapid Amplification) with (Fluorescence type) or without ExoIII (Basic type) (Amplification Future company, Weifang, Shandong Province, China) by FQ reporter and 5'-FAM-labeled ssDNA probe. All the reagents were added to the lyophilized powder following the usage instructions. Then 5 µl ssDNA FQ reporter or probe (10 µM) was added to the reaction mix. The tubes containing the reaction mixture were incubated in Mx3500P (Agilent, Santa Clara, CA, USA) at 37 °C for 30 min to monitor the fluorescence generated. The endpoint fluorescence was visualized under an LED blue light transilluminator (THBC-470) (TUOHE ELECTROMECHANICAL TECHNOLOGY CO., LTD, Shanghai, China).

## Acknowledgements

This work was supported by the Zhejiang Provincial Key R&D Program (2023C03045 & 2021C02008), the National Program on Key Research Project of China (2022YFC2604201 & 2019YFE0103900) as well as the European Union's Horizon 2020 Research and Innovation Programme under Grant Agreement No. 861917-SAFFI, Zhejiang Provincial Natural Science Foundation of China (LZ24C180002), Scientific Research Fund of Zhejiang Provincial Education Department (Y202250872), and Key Research and Development Program of Hangzhou (202203A08).

## Additional information

### Funding

| Funder | Grant reference number | Author |
|---|---|---|
| Zhejiang Provincial Key R&D Program | 2023C03045 | Min Yue |
| National Program on Key Research Project of China | 2022YFC2604201 | Min Yue |
| Horizon 2020 - Research and Innovation Framework Programme | 861917-SAFFI | Yan Li |

| Funder | Grant reference number | Author |
|---|---|---|
| Zhejiang Provincial Natural Science Foundation of China | LZ24C180002 | Min Yue |
| Scientific Research Fund of Zhejiang Provincial Education Department | Y202250872 | Min Yue |
| Key research and Development Program of Hangzhou | 202203A08 | Min Yue |
| National Program on Key Research Project of China | 2019YFE0103900 | Min Yue |
| Zhejiang Provincial Key R&D Program | 2021C02008 | Min Yue |

The funders had no role in study design, data collection, and interpretation, or the decision to submit the work for publication.

### Author contributions

Hao Wang, Conceptualization, Data curation, Software, Visualization, Methodology, Writing - original draft, Writing – review and editing; Chen Ye, Data curation, Formal analysis, Investigation, Visualization; Qi Lu, Na Li, Data curation, Investigation; Zhijie Jiang, Investigation, Methodology; Chao Jiang, Formal analysis, Validation, Methodology, Writing – review and editing; Chun Zhou, Formal analysis, Validation, Writing – review and editing; Caiqiao Zhang, Validation, Writing – review and editing; Guoping Zhao, Min Yue, Resources, Supervision, Writing – review and editing; Yan Li, Resources, Supervision, Validation, Writing – review and editing

### Author ORCIDs

Chun Zhou http://orcid.org/0000-0002-9257-468X
Caiqiao Zhang http://orcid.org/0000-0002-3519-9457
Min Yue https://orcid.org/0000-0002-6787-0794

Reviewer #2 (Public Review): https://doi.org/10.7554/eLife.95648.3.sa1
Reviewer #3 (Public Review): https://doi.org/10.7554/eLife.95648.3.sa2
Author response https://doi.org/10.7554/eLife.95648.3.sa3

# Additional files

### Supplementary files

- Supplementary file 1. All the reaction products of ExoIII obtained by mass spectrometry analysis are provided.
- Supplementary file 2. All the sequences used in the study are listed.
- MDAR checklist

### Data availability

All data generated or analysed during this study are included in the manuscript and supporting files.

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
