## [Editor Report · eLife assessment]

This manuscript highlights single-stranded DNA exo- and endo-nuclease activities of ExoIII as a potential caveat and an underestimated source of decreased efficiency in its use in biosensor assays. The data present **solid** evidence for the ssDNA nuclease activity of ExoIII and identifies residues that contribute to it. The findings are **useful**, but some aspects in the study remain **incomplete**.

---

## [Referee Report · Reviewer #2 (Public Review)]

Summary:

This paper describes some experiments addressing 3' exonuclease and 3' trimming activity of bacterial exonuclease III. The quantitative activity is in fact very low, despite claims to the contrary. The work is of low interest with regard to biology, but possibly of use for methods development. Thus the paper seems better suited to a methods forum.

Strengths:

Technical approaches.

Comments on revised version:

All concerns have been addressed.

---

## [Referee Report · Reviewer #3 (Public Review)]

Overall:

ExoIII has been described and commercialized as a dsDNA specific nuclease. Several lines of evidence, albeit incomplete, have indicated this may not be entirely true. Therefore, Wang et al comprehensively characterize the endonuclease and exonuclease enzymatic activities of ExoIII on ssDNA. A strength of the manuscript is the testing of popular kits that utilize ExoIII and coming up with and testing practical solutions (e.g., addition of SSB proteins ExoIII variants such as K121A and varied assay conditions).

Comments:

(1) The footprint of ExoIII on DNA is expected to be quite a bit larger than 5-nt, see structure in manuscript reference #5. Therefore, the substrate design in Figure 1A seems inappropriate for studying the enzymatic activity and it seems likely that ExoIII would be interacting with the FAM and/or BHQ1 ends as well as the DNA. Could this cause quenching? Would this represent real ssDNA activity? Is this figure/data necessary for the manuscript?

(2) Based on the descriptions in the text, it seems there is activity with some of the other nucleases in 1C, 1F, and 1I other than ExoIII and Cas12a. Can this be plotted on a scale that allows the reader to see these relative to one other?

(3) The sequence alignment in Figure 2N and corresponding text indicate a region of ExoIII lacking in APE1 that may be responsible for their differences in substrate specificity in regards to ssDNA. Does the mutational analysis support this hypothesis?

---

## [Author Response]

The following is the authors’ response to the original reviews.

**eLife assessment**
This manuscript highlights single-stranded DNA exo- and endo-nuclease activities of ExoIII as a potential caveat and an underestimated source of decreased efficiency in its use in biosensor assays. The data present convincing evidence for the ssDNA nuclease activity of ExoIII and identifies residues that contribute to it. The findings are useful, but the study remains incomplete as the effect on biosensor assays was not established.
**Reviewer #1 (Public Review):**
Summary:In this manuscript, the authors show compelling data indicating that ExoIII has significant ssDNA nuclease activity that is posited to interfere with biosensor assays. This does not come as a surprise as other published works have indeed shown the same, but in this work, the authors provide a deeper analysis of this underestimated activity.

Response: Thank you so much for reviewing and summarizing our work.

Strengths:The authors used a variety of assays to examine the ssDNA nuclease activity of ExoIII and its origin. Fluorescence-based assays and native gel electrophoresis, combined with MS analysis clearly indicate that both commercial and laboratory purified ExoIII contain ssDNA nuclease activity. Mutational analysis identifies the residues responsible for this activity. Of note is the observation in this submitted work that the sites of ssDNA and dsDNA exonuclease activity overlap, suggesting that it may be difficult to identify mutations that affect one activity but not the other. In this regard, it is of interest the observation by the authors that the ssDNA nuclease activity depends on the sequence composition of the ssDNA, and this may be used as a strategy to suppress this activity when necessary. For example, the authors point out that a 3′ A4-protruding ssDNA could be employed in ExoIII-based assays due to its resistance to digestion. However, this remains an interesting suggestion that the authors do not test, but that would have strengthened their conclusion.

Response: Thank you so much for the positive evaluation and insightful comments on our manuscript. In the revised version, we have modified the manuscript to address the reviewer’s concerns by providing point-to-point responses to all the comments.

Weaknesses:The authors provide a wealth of experimental data showing that *E. coli* ExoIII has ssDNA nuclease activities, both exo- and endo-, however this work falls short in showing that indeed this activity practically interferes with ExoIII-driven biosensor assays, as suggested by the authors. Furthermore, it is not clear what new information is gained compared to the one already gathered in previously published works (e.g. references 20 and 21). Also, the authors show that ssDNA nuclease activity has sequence dependence, but in the context of the observation that this activity is driven by the same site as dsDNA Exo, how does this differ from similar sequence effects observed for the dsDNA Exo? (e.g. see Linxweiler, W. and Horz, W. (1982). Nucl. Acids Res. 10, 4845-4859).

Response: We agree with the reviewer regarding the limitations in showing the practical influence of the ssDNAse activity in the commercial detection kit. Different from the biosensor in reference 20, our results showed a potential impact of ExoⅢ on another frequently used detection system, as the primer and probe required for the detection kit could be digested by ExoⅢ, leading to a lower detection efficiency. Since the activities of ExoⅢ on ssDNA and dsDNA share a same active center, we reason that the difference in sequence specificity of ExoⅢ on these two types of substrates might be caused in two aspects: on the nuclease, some unidentified residues of ExoⅢ that play an auxiliary role in digesting ssDNA but not in dsDNA, might exist, which contribute to the difference we observed; on the substrate structure, without the base-pairing of complementary sequence, the structure of ssDNA is more flexible (changeable with environmental factors such as ions and temperature) than that of dsDNA. The two aspects may collectively result in the difference in sequence specificity of ExoⅢ on ssDNA and dsDNA. We believe that cryo-electronic microscopy-based structure analysis of the ExoⅢ-ssDNA complex would provide more comprehensive and direct evidence.

Because of the claim that the underestimated ssDNA nuclease activity can interfere with commercially available assays, it would have been appropriate to test this. The authors only show that ssDNA activity can be identified in commercial ExoIII-based kits, but they do not assess how this affects the efficiency of a full reaction of the kit. This could have been achieved by exploiting the observed ssDNA sequence dependence of the nuclease activity. In this regard, the work cited in Ref. 20 showed that indeed ExoIII has ssDNA nuclease activity at concentrations as low as 50-fold less than what test in this work. Ref 20 also tested the effect of the ssDNA nuclease activity in Targeted Recycle Assays, rather than just testing for its presence in a kit.

Response: Thanks so much for your comments. Logically, to evaluate the practical influence, we need to compare the current and improved detection kits. Our result suggested that raising the temperature or using the mutant may minimize the ssDNase activity of ExoⅢ. But the RAA or RPA-ExoⅢ detection kit is multiple-component system consisting of recombinase T4 UvsX, loading factor T4 UvsY, ssDNA binding protein T4 gp32 polymerase Bsu and ExoⅢ (Analyst. 2018 Dec 17;144(1):31-67. doi: 10.1039/c8an01621f), which collectively decide the performance of the kit. By increasing the temperature, the activities or functions of other proteins contained in the detection kit would also be affected, and the resultant change in detection efficiency would not reflect the real practical influence of the ssDNase activity of ExoⅢ; By replacing the wild type with the mutant, the other four proteins need to be prepared and combined with an optimized ratio for rebuilding the detection system, which is challenging. The targeted recycle assays in Ref 20 is a simple system composed of ExoⅢ and corresponding nucleic acid adapters, which could be easily simulated by the researchers for evaluation. Being a much more complex system, the RAA or RPA-ExoⅢ detection kit is difficult to manipulate for displaying the practical influence. Thank you again for your insightful suggestions; and we may conduct a systematic investigation improve the detection kit in future studies.

Because of the implication that the presence of ssDNA exonuclease activity may have in reactions that are supposed to only use ExoIII dsDNA exonuclease, it is surprising that in this submitted work no direct comparison of these two activities is done. Please provide an experimental determination of how different the specific activities for ssDNA and dsDNA are.

Response: As for your suggestion, we have compared the digesting rate of two activities by using an equal amount of the commercial ExoⅢ (10 U/µL) on the two types of substrates (10 µM). The results below revealed that ExoⅢ required 10 minutes to digest the 30-nt single-stranded DNA (ssDNA) (A), whereas it could digest the same sequence on double-stranded DNA (dsDNA) within 1 minute (B) (in a newly produced Supplementary Figure S1). This indicated that ExoⅢ digested the dsDNA at a rate at least ten times faster than ssDNA. In conjunction with these results, a recent study has shown that the ssDNase activity of ExoⅢ surpasses that of the conventional ssDNA-specific nuclease ExoI (Biosensors (Basel), 2023, May 26; 13(6):581, doi: 10.3390/bios13060581), suggesting a potential biological significance of ExoⅢ in bacteria related to ssDNA, even though the digesting rate is not as rapid as the dsDNA. The corresponding text has been added to the result (Lines 200-207).

**Reviewer #2 (Public Review):**
Summary:This paper describes some experiments addressing 3' exonuclease and 3' trimming activity of bacterial exonuclease III. The quantitative activity is in fact very low, despite claims to the contrary. The work is of low interest with regard to biology, but possibly of use for methods development. Thus the paper seems better suited to a methods forum.

Response: We thank you for your time and effort in improving our work. In the following, we have revised the manuscript by providing point-to-point responses to your comments.

Strengths:Technical approaches.

Response: Thanks for your evaluation.

Weaknesses:The purity of the recombinant proteins is critical, but no information on that is provided. The minimum would be silver-stained SDS-PAGE gels, with some samples overloaded in order to detect contaminants.

Response: As suggested, we have performed the silver-stained SDS-PAGE on the purified proteins. The result below indicated that no significant contaminant was found, except for a minor contaminant in S217A (in a newly produced Supplementary Figure S4).

**Author response image 2. sa3fig2:** 

Lines 74-76: What is the evidence that BER in *E. coli* generates multinucleotide repair patches in vivo? In principle, there is no need for the nick to be widened to a gap, as DNA Pol I acts efficiently from a nick. And what would control the extent of the 3' excision?

Response: Thank you for the insightful questions. The team of Gwangrog Lee lab has found that ExoⅢ is capable of creating a single-stranded DNA (ssDNA) gap on dsDNA during base excision repair, followed by the repair of DNA polymerase I. The gap size is decided by the rigidity of the generated ssDNA loop and the duplex stability of the dsDNA (Sci Adv. 2021 Jul 14;7(29):eabg0076. doi: 10.1126/sciadv.abg0076).

Figure 1: The substrates all report only the first phosphodiester cleavage near the 3' end, which is quite a limitation. Do the reported values reflect only the single phosphodiester cleavage? Including the several other nucleotides likely inflates that activity value. And how much is a unit of activity in terms of actual protein concentration? Without that, it's hard to compare the observed activities to the many published studies. As best I know, Exo III was already known to remove a single-nucleotide 3'-overhang, albeit more slowly than the digestion of a duplex, but not zero! We need to be able to calculate an actual specific activity: pmol/min per µg of protein.

Response: Yes, once the FQ reporter is digested off even one nucleotide or phosphodiester, fluorescence will be generated, and the value reflects how many phosphodiesters at least have been cleaved during the period, based on which the digesting rate or efficiency of the nuclease on ssDNA could be calculated. The following Figure 2 and 3 showed ExoⅢ could digest the ssDNA from the 3’ end, not just a single nucleotide. Since the “unit” has been widely used in numerous studies (Nature. 2015 Sep 10;525(7568):274-7; Cell. 2021 Aug 19;184(17):4392-4400.e4; Nat Nanotechnol. 2018 Jan;13(1):34-40.), its inclusion here aids in facilitating comparisons and evaluations of the activity in these studies. And the actual activity of ExoⅢ had been calculated in Figure 4D.

Figures 2 & 3: These address the possible issue of 1-nt excision noted above. However, the question of efficiency is still not addressed in the absence of a more quantitative approach, not just "units" from the supplier's label. Moreover, it is quite common that commercial enzyme preparations contain a lot of inactive material.

Response: Thanks for your comments. In fact, numerous studies have used the commercial ExoⅢ (Nature. 2015 Sep 10;525(7568):274-7; Cell. 2021 Aug 19;184(17):4392-4400.e4; Nat Nanotechnol. 2018 Jan;13(1):34-40.). Using this universal label of “units” helps researchers easily compare or evaluate the activity and its influence. The commercial ExoⅢ is developed by New England Biolabs Co., Ltd., and its quality has been widely examined in a wide range of scientific investigations.

Figure 4D: This gets to the quantitative point. In this panel, we see that around 0.5 pmol/min of product is produced by 0.025 µmol = 25,000 pmol of the enzyme. That is certainly not very efficient, compared to the digestion of dsDNA or cleavage of an abasic site. It's hard to see that as significant.

Response: Thanks for your comments; the possible confusion could have arisen due to the arrangement of the figure. Please note that based on Figure 4D, the digestion rate of 0.025 µM ExoⅢ on the substrate is approximately 5 pmol/min (as shown on the right vertical axis), rather than 0.5 pmol/min. Given that the reaction contained ExoⅢ with a concentration of 0.025 uM in a total volume of 10 µL, the quantity of ExoⅢ was determined to be 0.25 pmol (0.025 µmol/L × 10 µL, rather than 25,000 pmol), resulting in a digestion rate of 5 pmol/min. It suggested each molecule of ExoⅢ could digest one nucleotide in 3 seconds (5 pmol nucleotides /0.25 pmol ExoⅢ/60second=0.33 nucleotides/molecular/second). While it may not be as rapid as the digestion of ExoⅢ on dsDNA, a recent study has shown that the ssDNase activity of ExoⅢ surpasses that of the conventional ssDNA-specific nuclease ExoI (Biosensors (Basel), 2023, May 26; 13(6):581, doi: 10.3390/bios13060581), suggesting a potential biological significance of ExoⅢ in bacteria related to ssDNA.

Line 459 and elsewhere: as noted above, the activity is not "highly efficient". I would say that it is not efficient at all.

Response: We respectfully disagree with this point. Supported by the outcomes from fluorescence monitoring of FQ reporters, gel analysis of the ssDNA probe, and mass spectrometry findings, the conclusion is convincing, and more importantly, our findings align with a recent study (Biosensors 2023, 13(6), 581; https://doi.org/10.3390/bios13060581).

**Reviewer #3 (Public Review):**
Overall:ExoIII has been described and commercialized as a dsDNA-specific nuclease. Several lines of evidence, albeit incomplete, have indicated this may not be entirely true. Therefore, Wang et al comprehensively characterize the endonuclease and exonuclease enzymatic activities of ExoIII on ssDNA. A strength of the manuscript is the testing of popular kits that utilize ExoIII and coming up with and testing practical solutions (e.g. the addition of SSB proteins ExoIII variants such as K121A and varied assay conditions).

Response: We really appreciate the reviewer for pointing out the significance and strength of our work. Additionally, we have responded point-by-point to the comments and suggestions.

Comments:(1) The footprint of ExoIII on DNA is expected to be quite a bit larger than 5-nt, see structure in manuscript reference #5. Therefore, the substrate design in Figure 1A seems inappropriate for studying the enzymatic activity and it seems likely that ExoIII would be interacting with the FAM and/or BHQ1 ends as well as the DNA. Could this cause quenching? Would this represent real ssDNA activity? Is this figure/data necessary for the manuscript?

Response: Thanks so much for your questions. The footprint of ExoⅢ on the dsDNA appears to exceed 5 nucleotides based on the structural analysis in reference #5. However, the footprint may vary when targeting ssDNA. Mass spectrometry analysis in our study demonstrated that ExoⅢ degraded a ~20-nucleotide single-stranded DNA substrate to mononucleotides (Figure 3), suggesting its capability to digest a 5-nt single-stranded DNA into mononucleotides as well. Otherwise, the reaction product left would only be 5-nt ssDNA fragment. Thus, the 5-nt FQ reporter is also a substrate for ExoⅢ. ExoⅢ possibly interacts with BHQ1 and affects its quenching efficiency on FAM to trigger the fluorescence release, as shown in Figure 1A, but this possibility has already been ruled out by the development of the RPA-ExoⅢ detection kit. As pointed out in the introduction part, the kit requires a probe labeled with fluorophore and quencher. If ExoⅢ could affect the fluorophore and quencher causing fluorescence release, the detection kit would yield a false-positive result regardless of the presence of the target, rendering the detection system ineffective. Thus, ExoⅢ does not interfere with the fluorophore and quencher. The digestion of ExoⅢ on the ssDNA within the FQ reporter was the sole cause of fluorescence release, and the emitted fluorescence represented the ssDNA activity. The result suggested that the FQ reporter might offer an effective approach to sensitively detect or quantitatively study the ssDNase activity of a protein that has not been characterized.

(2) Based on the descriptions in the text, it seems there is activity with some of the other nucleases in 1C, 1F, and 1I other than ExoIII and Cas12a. Can this be plotted on a scale that allows the reader to see them relative to one other?

Response: Thanks so much for your suggestions. We attempted to adjust the figure, but due to most of the values being less than or around 0.005, it was challenging to re-arrange for presentation.

(3) The sequence alignment in Figure 2N and the corresponding text indicates a region of ExoIII lacking in APE1 that may be responsible for their differences in substrate specificity in regards to ssDNA. Does the mutational analysis support this hypothesis?

Response: Our result indicated that the mutation of R170 located in the region (αM helix) resulted in lower digesting efficiency on ssDNA than the wild type, which showed that R170 was an important residue for the ssDNase activity, partially supported the hypothesis. Further investigation is needed to determine whether the structure of the αM helix accounts for the distinctions observed between ExoⅢ and APE1. Future research may require more residue mutations in this area for validation.

**Recommendations for the authors:**

**Reviewer #1 (Recommendations For The Authors):**
A significant fraction of amplitude is missing in the presented fluorescence time courses reporting on ssDNA nuclease activity (Figs 1 B, E, and H). Please indicate the dead time of mixing in these experiments, and if necessary include additional points in this time scale. It is unacceptable for the authors to simply connect the zero-time point and the first experimental point with a dashed line.

Response: We thank the reviewer for pointing out the critical detail. We agree that simply connecting with a dashed line is an inappropriate way for indicating the real fluorescence generated in the initial stage. The fluorescence monitor machine needs about two minutes to initiate from the moment we place the reaction tube into the machine. But ExoⅢ can induce significant fluorescence immediately, reaching the peak within ~40 seconds, as shown in the video data. Therefore, it is difficult to record the initial real-time fluorescence generated. To avoid misleading, we have added a description in the legend as follows: “The dashed line used in the figure does not indicate the real-time fluorescence generated in the reaction but only represents a trend in the period for the monitor machine to initiate (~2 minutes).” The text was added in Lines 836-838.

The authors chose to utilize a 6% agarose electrophoresis to analyze digestion products. However, while this approach clearly shows that the substrates are being digested, it does not allow us to clearly estimate the extent. It would be appropriate to include control denaturing PAGE assays to test the extent of reaction, especially for dsDNA that contains a ssDNA extension, as in Figure 8, or for selected mutants to test whether exo activity may be limited to just a few nts, that may not be resolved with the lower resolution agarose gels.

Response: We agree with the reviewer that denaturing PAGE assays usually is the choice for high-resolution analysis. And we performed this experiment on the short ssDNA, but observed that the bands of digestion products frequently shifted more or less in the gel. Of note, the other independent study also showed a similar phenomenon (Nucleic Acids Res. 2007;35(9):3118-27. doi: 10.1093/nar/gkm168). Even slight band shifting would significantly interfere with our analysis of the results, especially on the short ssDNA utilized in the study. After numerous attempts, we discovered that 6% agarose gel electrophoresis could detect the digested ssDNA bands with lower resolution than PAGE, but less shifting was observed. Considering all the factors, the 6% agarose gel was finally selected to analyze the digestion process.

**Reviewer #2 (Recommendations For The Authors):**
Line 158: tipycal should be typical

Response: Thanks so much, and as the reviewer pointed, we have corrected the typo.

Lines 299-300: "ssD-NA" should not be hyphenated, i.e., it should be ssDNA. .

Response: Thank you for pointing this out. We have rectified the error and thoroughly reviewed the entire paper for any necessary corrections.

**Reviewer #3 (Recommendations For The Authors):**
Figure 2A should indicate the length of the substate. The legend says omitted nucleotides - I assume they were present in the substrate and just not in the figure? The authors should be very clear about this. Moreover, the text and figure do not well describe the design differences between the three probes. Are they the same except just 23, 21, and 20 nt in length? Are the sequences selected at random?

Response: Thank you for your questions. The lengths of probes were described in the figure (23, 21, and 20 nt). The legend has been reworded in Line 843 as “The squiggle line represents the ~20 nucleotides of the ssDNA oligo.” And the sequences of three ssDNA substrates were randomly selected, and all the detailed information was provided in Supplementary Table S4.